



# BFM17 v1.0: Reduced-Order Biogeochemical Flux Model for Upper Ocean Biophysical Simulations

Katherine M. Smith[1], Skyler Kern[1], Peter E. Hamlington[1], Marco Zavatarelli[2], Nadia Pinardi[2], Emily F. Klee[3], and Kyle E. Niemeyer[3]

[1] Paul M. Rady Department of Mechanical Engineering, University of Colorado, Boulder, CO, USA

[2] Department of Physics and Astronomy, University of Bologna, Bologna, IT

[3] School of Mechanical, Industrial, and Manufacturing Engineering, Oregon State University, Corvallis, OR, USA

**Correspondence:** P. E. Hamlington (peh@colorado.edu)

**Abstract.** We present a newly developed reduced-order biogeochemical flux model that is complex and flexible enough to capture open-ocean ecosystem dynamics, but reduced enough to incorporate into highly resolved numerical simulations with limited additional computational cost. The reduced-order model, which is derived from the full 56 state variable Biogeochemical Flux Model (BFM56; Vichi et al. (2007)), follows a biological and chemical functional group approach and allows for the

development of critical non-Redfield nutrient ratios. Matter is expressed in units of carbon, nitrogen, and phosphate, following techniques used in more complex models. To reduce the overall computational cost and to focus on open-ocean conditions, the reduced model eliminates certain processes, such as benthic, silicate, and iron influences, and parameterizes others, such as the bacterial loop. The model explicitly tracks 17 state variables, divided into phytoplankton, zooplankton, dissolved organic matter, particulate organic matter, and nutrient groups. It is correspondingly called the Biogeochemical Flux Model 17 (BFM17).

After providing a detailed description of BFM17, we couple it with the one-dimensional Princeton Ocean Model (POM) for validation using observational data from the Sargasso Sea. Results show good agreement with the observational data and with corresponding results from BFM56, including the ability to capture the subsurface chlorophyll maximum and bloom intensity. In comparison to previous reduced-order models of similar size, BFM17 provides improved correlations between model output and field data, indicating that significant improvements in the reproduction of *in situ* data can be achieved with a low number

of variables, while maintaining the functional group approach.

## 1  Introduction

Biogeochemical (BGC) tracers and their interactions with upper-ocean physical processes, from basin-scale circulations to millimeter-scale turbulent dissipation, are critical for understanding the role of the ocean in the global carbon cycle. These interactions cause multi-scale spatial and temporal heterogeneity in tracer distributions (Strass, 1992; Yoder et al., 1992; Jr.

et al., 2001; Gower et al., 1980; Denman and Abbott, 1994; Strutton et al., 2012; Clayton, 2013; Abraham, 1998; Bees, 1998; Mahadevan and Archer, 2000; Mahadevan and Campbell, 2002; Levy and Klein, 2015; Powell and Okubo, 1994; Martin et al., 2002; Mahadevan, 2005; Tzella and Haynes, 2007) that can greatly affect carbon exchange rates between the atmosphere and interior ocean, net primary productivity, and carbon export (Lima et al., 2002; Schneider et al., 2007; Hauri et al., 2013;



Behrenfeld, 2014; Barton et al., 2015; Boyd et al., 2016). There are still significant gaps, however, in our understanding of how
these biophysical interactions develop and evolve, thus limiting our ability to accurately predict critical exchange rates.

Better understanding these interactions requires accurate physical and BGC models that can be coupled together. The exact
equations that describe the physics (e.g., the Navier–Stokes or Boussinesq equations) are often known and physically accurate
solutions can be obtained given sufficient spatial resolution and computational resources. Due to the vast diversity and com-
plexity of ocean ecology, however, even when only considering the lowest trophic levels, accurately modeling BGC processes
can be quite difficult. Put simply, there are no known first-principles governing equations for ocean biology.

As such, two different approaches to modeling BGC processes are often used when faced with this challenge. The first is to
increase model complexity and include equations for every known BGC process. Often, these models include species functional
types or multiple classes of phytoplankton and/or zooplankton that each serve specific functional roles within the ecosystem,
such as calcifiers or nitrogen fixers. The justification for this approach is that particular phytoplankton and zooplankton groups
serve as important system feedback pathways, and that without explicit representation of these feedbacks, there is little hope of
accurately representing the target ecosystem (Doney, 1999; Anderson, 2005). In many cases, these models also contain variable
intra- and extra-cellular nutrient ratios, which are important when accounting for different nutrient regimes within the global
ocean and species diversity of non-Redfield nutrient ratio uptake (Dearman et al., 2003).

Although these more complex models are typically highly adaptable and are often able to capture vastly different dynamics
than those for which they were calibrated (Blackford et al., 2004; Friedrichs et al., 2007), they contain many more parameters
than their simplified counterparts. Moreover, many of the parameters, such as phytoplankton mortality, zooplankton grazing
rates, and bacterial remineralization rates, are inadequately bounded by either observational or experimental data (Denman,
2003). Because of the increased complexity of such models, it is also often difficult to ascertain which processes are responsible
for the development of a particular event (e.g., a phytoplankton bloom), and so these models can be ill-suited for process studies.
Lastly, while these highly complex models are regularly used within global Earth System Models (ESMs), they are typically
prohibitively expensive to integrate within high-fidelity, high-resolution physical models at submesoscales, such as those used
to enhance fundamental understanding of subgrid-scale (SGS) physics in ESMs and to assist in the development of new SGS
parameterizations (Roekel et al., 2012; Hamlington et al., 2014; Suzuki and Fox-Kemper, 2015; Smith et al., 2016, 2018).

In broad terms, the second BGC modeling approach is focused on substantially decreasing model complexity and severely
truncating the number of equations used to describe the dynamics of an ecosystem. Such approaches include the well-known
nutrient-phytoplankton-zooplankton-detritus class of models. These models have significantly fewer unknown parameters and
can be more easily integrated within complex physical models. Their simplicity also enables greater transparency when attempt-
ing to understand the dominant forcing or dynamics underlying a particular event. While they are often capable of reproducing
the overall distributions of chlorophyll, primary production, and nutrients (Anderson, 2005), such simplified models have been
shown to under-perform at capturing complex ecosystem dynamics, and often struggle in regions of the ocean for which they
were not calibrated (Friedrichs et al., 2007).

Although both of these general BGC modeling approaches have their respective advantages, particularly given their vastly
different objectives, the disconnect between reduced-order BGC models used in small-scale studies and the more complex



BGC models used in global ESMs poses a problem. In particular, the difficulty in directly comparing the two types of models
makes the process of "scaling-up" newly developed parameterizations or "downscaling" BGC variables within nested-grid
studies much more challenging. This motivates the need for a new BGC model that is reduced enough to be usable within
high-resolution, high-fidelity physical simulations for process studies and parameterization development, but is still complex
enough to capture important ecosystem feedback dynamics, as well as the dynamics of vastly different ecosystems throughout
the ocean, as required by ESMs.

To begin addressing this need, here we present a new reduced-order, 17 state-variable Biogeochemical Flux Model (BFM17)
obtained by reducing the larger 56 state variable Biogeochemical Flux Model (BFM56) developed by Vichi et al. (2007). Most
high-fidelity, high-resolution physical models are capable of integrating 17 additional tracer equations with limited additional
computational cost. Following the approach used in BFM56 (Vichi et al., 2007, 2013), a biological and chemical functional
family (CFF) approach underlies BFM17, permitting variable non-Redfield intra- and extra-cellular nutrient ratios, and matter is
exchanged in the model through units of carbon, nitrate, and phosphate. Most notably, BFM17 includes a phosphate budget, the
importance of which has historically been under-appreciated even though recent observational data has indicated its potential
importance as a limiting nutrient, particularly in the Atlantic Ocean (Ammerman et al., 2003). To reduce model complexity,
we parameterize certain processes for which field data are lacking, such as bacterial remineralization.

In the present study, we outline, in detail, the formulation of BFM17 and its development from BFM56. We couple BFM17
to the one-dimensional Princeton Ocean Model (POM) and validate the model for open-ocean conditions using observational
data from the Sargasso Sea. We also compare results from BFM17 and the larger BFM56 for the same open-ocean conditions.
As a result of the focus on open-ocean conditions, further assumptions have been made in obtaining BFM17 from BFM56,
such as the exclusion of any representation for the benthic system and the absence of limiting nutrients such as iron and silicate.

It should be noted that the primary focus in the present study is to demonstrate the accuracy of BFM17 when compared
to results from observations and BFM56; as such, here we only consider one open-ocean location (i.e., the Sargasso Sea).
However, the correspondence between BFM17 and the more general BFM56 provides confidence that the reduced model will
also prove effective at modeling other ocean locations and conditions, and exploring the range of applicability of BFM17
remains a subject of future research. We also emphasize that relatively limited calibration of BFM17 parameters has been
performed in the present study. Most parameters are set to their values used in the larger BFM56 (Vichi et al., 2007, 2013), and
optimization of these parameters over a range of ocean conditions is another important direction of future research.

Finally, we note that other reduced-order BGC models have been calibrated using data from the Sargasso Sea, such as those
developed in Levy et al. (2005), Ayata et al. (2013), Spitz et al. (2001), Doney et al. (1996), Fasham et al. (1990), Fennel
et al. (2001), Hurtt and Armstrong (1996), Hurtt and Armstrong (1999), and Lawson et al. (1996). However, each of these
models employs less than 10 species and none uses a CFF approach. Although some of these models employ data assimilation
techniques (e.g., Spitz et al. (2001)) and produce relatively accurate results, most leave room for improvement. With a minimal
increase in the number and complexity of the model equations, such as those associated with tracking phosphate in addition
to carbon and nitrate, and including both particulate and dissolved organic nutrient budgets, we postulate that a significant
increase in model accuracy can be achieved over previous reduced-order models. Additionally, with this increase in model





complexity, the disparate gap between the complexity of BGC models used in small- and global-scale studies is reduced,
thereby simplifying up- and down-scaling efforts. This last point is emphasized here by the good agreement between results
from BFM17 and BFM56.

In the following, the zero-dimensional (0D) BFM17 model is introduced in Section 2. In Section 3, BFM17 is coupled to the
one-dimensional (1D) POM physical model. A discussion of the methods used to calibrate and validate the model with field
data collected within the Sargasso Sea is presented in Section 4. Model results, a skill assessment, a comparison to results from
BFM56, and a brief comparison to other similar BGC models are discussed in Section 5.

## 2  Biogeochemical Flux Model 17 (BFM17)

The 17 state equation BFM17 is a reduced-order BGC model derived from the original 56 state equation BFM56 (Vichi et al.,
2007, 2013), which is based on the CFF approach. In this approach, functional groups are partitioned into living organic, non-
living organic, and non-living inorganic CFFs, and exchange of matter occurs through constituent units of carbon, nitrogen,
and phosphate. To date, there are no other BGC models with this order of reduced complexity using the CFF approach, making
BFM17 unique and able to accurately reproduce complex ecosystem dynamics.

The reduced-order BFM17 is a pelagic model intended for upper thermocline, open-ocean, oligotrophic regions and is
obtained from the more-complete BFM56 by omitting quantities and processes of lesser significance in these regions, subject
to the constraint that variable internal nutrient dynamics are of continued importance. In BFM17, the living organic CFF is
comprised of single phytoplankton and zooplankton living functional groups (LFGs); these two groups are the bare minimum
needed within a BGC model and already account for six state equations (corresponding to carbon, nitrogen, and phosphate
constituents of both groups). The baseline parameters used to model phytoplankton loosely correspond to the flagellate LFG
in BFM56, while the zooplankton parameters correspond to the micro-zooplankton LFG (Vichi et al., 2007, 2013). Compared
to BFM56, some of the parameter values in BFM17 were altered to represent general phytoplankton and zooplankton LFGs
and to improve agreement with observational data, although most parameters retain the same values as in BFM56. Dissolved
and particulate organic matter, each with their own partitions of carbon, nitrogen, and phosphate, are also included to account
for nutrient recycling and carbon export due to particle sinking, both of which are important in upper thermocline, open-ocean,
oligotrophic regions. Remineralization of nutrients is provided by parameterized bacterial closure terms, thereby reducing
complexity while still maintaining critical nutrient recycling. Lastly, we track chlorophyll, dissolved oxygen, phosphate, nitrate,
and ammonium, since their distributions and availability can greatly enhance or hinder important biological and chemical
processes.

Iron is omitted from BFM17, limiting the applicability of the model in regions where iron components are important, such as
the Southern Ocean. Top-down control of the ecosystem is also not included. Instead, a simple constant zooplankton mortality
is used, as this is a complicated process and understanding where to add this closure and where to feed the particulate and
dissolved nutrients from this process in a reduced-order model is not well understood. However, the addition of a top-down
closure term was tested in various ways and no major differences were observed in the model results. Consequently, it was





assumed that the constant mortality term was sufficient for this model, similar to other reduced-order models (Fasham et al., 1990; Lawson et al., 1996; Clainche et al., 2004). Additionally, the benthic system within BFM56 (Mussap et al., 2016) has been removed. It is assumed that within the upper thermocline of the open ocean, the ecosystem is not substantially influenced

by a benthic system and any water-column influences from depth can be taken into account using boundary conditions (such as those discussed in Section 4).

In summary, notable novel attributes of BFM17, in comparison with other reduced-order models of comparable complexity, are the use of (*i*) CFFs for living organisms, including two LFGs for phytoplankton and zooplankton, (*ii*) CFFs for both particulate and dissolved organic matter, and (*iii*) a full nutrient profile (i.e., phosphate, nitrate, and ammonium).

## 2.1 BFM17 Model Equations

In the following, the detailed equations for each of the 17 state variables that comprise BFM17 are outlined. A summary of the 17 state variables is provided in Table 1 and a schematic of the CFFs and LFGs used in BFM17, along with their interactions, is shown in Figure 1.

### 2.1.1 Environmental parameters

The BFM17 interacts with the environment through temperature and irradiance inputs. Temperature directly affects all physiological processes and is represented in the model by introducing the non-dimensional parameter $f_j^{(T)}$ defined as

$$f_j^{(T)} = Q_{10,j}^{(T-T^*)/T^*}, \quad j = P, Z, \tag{1}$$

where $T^*$ is a base temperature and $Q_{10,j}$ is a coefficient that may differ for the phytoplankton and zooplankton LFGs, denoted $P_i$ and $Z_i$, respectively. Here, the subscript $i$ is used to denote different chemical constituents (i.e., C, N, and P) and $j$ is

used to denote different LFGs. Base values used for $T^*$ and $Q_{10,j}$ are shown in Table 2. The model additionally employs a temperature-dependent nitrification parameter $f_N^{(T)}$, which is defined similarly to Eq. (1) as

$$f_N^{(T)} = Q_{10,N}^{(T-T^*)/T^*}, \tag{2}$$

where $Q_{10,N}$ is given in Table 2

In contrast to temperature, irradiance only directly affects phytoplankton, serving as the primary energy source for phyto-

plankton growth and maintenance. Irradiance is a function of the incident solar radiation at the sea surface. Within BFM17, the amount of photosynthetically active radiation (PAR) at any given location $z$ is parameterized according to the Lambert–Beer model as

$$E_{PAR}(z) = \varepsilon_{PAR} Q_S \exp\left[\lambda_w z + \int_z^0 \lambda_{bio}(z')dz'\right], \tag{3}$$

where $Q_S$ is the short-wave surface irradiance flux, which is typically obtained from real-world measurements of the atmo-

spheric radiative transfer, $\varepsilon_{PAR}$ is the fraction of PAR within $Q_S$, $\lambda_w$ is the background light extinction due to water, and $\lambda_{bio}$ is





**Table 1.** Notation used for the 17 state variables in the BFM17 pelagic model, as well as the chemical functional family (CFF), units, description, and rate equation reference for each state variable. CFFs are divided into living organic (LO), non-living organic (NO), and inorganic (IO) families.

| Symbol | CFF | Units | Description | Equation |
|--------|-----|-------|-------------|----------|
| $P_C$ | LO | mg C m$^{-3}$ | Phytoplankton carbon | (5) |
| $P_N$ | LO | mmol N m$^{-3}$ | Phytoplankton nitrogen | (6) |
| $P_P$ | LO | mmol P m$^{-3}$ | Phytoplankton phosphorus | (7) |
| $P_{chl}$ | LO | mg Chl-$a$ m$^{-3}$ | Phytoplankton chlorophyll | (8) |
| $Z_C$ | LO | mg C m$^{-3}$ | Zooplankton carbon | (31) |
| $Z_N$ | LO | mmol N m$^{-3}$ | Zooplankton nitrogen | (32) |
| $Z_P$ | LO | mmol P m$^{-3}$ | Zooplankton phosphorus | (33) |
| $R_C^{(1)}$ | NO | mg C m$^{-3}$ | Dissolved organic carbon | (41) |
| $R_N^{(1)}$ | NO | mmol N m$^{-3}$ | Dissolved organic nitrogen | (42) |
| $R_P^{(1)}$ | NO | mmol P m$^{-3}$ | Dissolved organic phosphorus | (43) |
| $R_C^{(2)}$ | NO | mg C m$^{-3}$ | Particulate organic carbon | (44) |
| $R_N^{(2)}$ | NO | mmol N m$^{-3}$ | Particulate organic nitrogen | (45) |
| $R_P^{(2)}$ | NO | mmol P m$^{-3}$ | Particulate organic phosphorus | (46) |
| $O$ | IO | mmol O$_2$ m$^{-3}$ | Dissolved oxygen | (47) |
| $N^{(1)}$ | IO | mmol P m$^{-3}$ | Phosphate | (48) |
| $N^{(2)}$ | IO | mmol N m$^{-3}$ | Nitrate | (49) |
| $N^{(3)}$ | IO | mmol N m$^{-3}$ | Ammonium | (50) |

**Table 2.** Symbols, values, units, and descriptions for environmental parameters within the BFM17 pelagic model.

| Symbol | Value | Units | Description |
|--------|-------|-------|-------------|
| $Q_{10,P}$ | 2.00 | - | Phytoplankton $Q_{10}$ coefficient |
| $Q_{10,Z}$ | 2.00 | - | Zooplankton $Q_{10}$ coefficient |
| $Q_{10,N}$ | 2.00 | - | Nitrification $Q_{10}$ coefficient |
| $T^*$ | 10.0 | °C | Base temperature |
| $c_P$ | 0.03 | m$^2$ (mg chl)$^{-1}$ | Chlorophyll-specific light absorption coefficient |
| $\varepsilon_{PAR}$ | 0.40 | - | Fraction of photosynthetically active radiation |
| $\lambda_w$ | 0.0435 | m$^{-1}$ | Background attenuation coefficient |
| $c_{R^{(2)}}$ | $0.1 \times 10^{-3}$ | m$^2$ (mg C)$^{-1}$ | C-specific attenuation coefficient of particulate detritus |





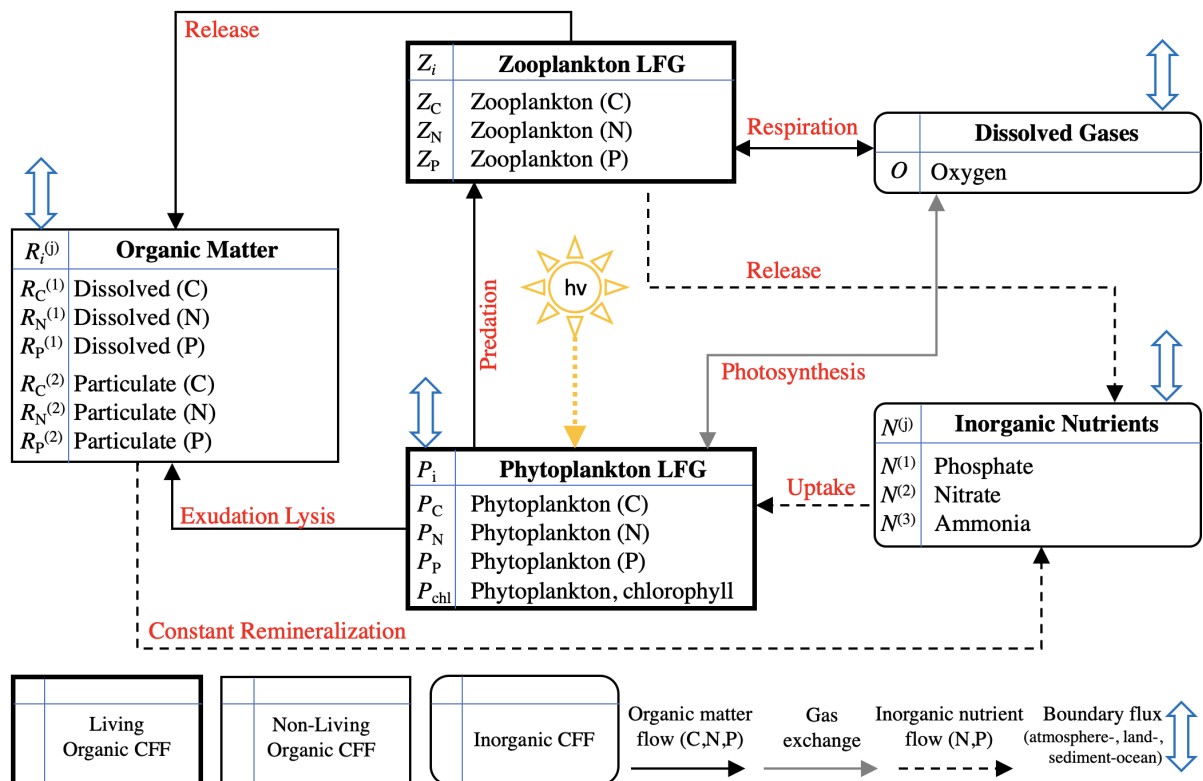

**Figure 1.** Schematic of the 17 state equation BFM17 pelagic model. The dissolved organic matter, particulate organic matter, and living organic matter chemical functional families (CFFs) are each comprised of three chemical constituents (i.e., carbon, nitrogen, and phosphorus). The living organic CFF is further subdivided into phytoplankton and zooplankton living functional groups (LFGs).

the light extinction due to suspended biological particles. Values for $\varepsilon_{\mathrm{PAR}}$ and $\lambda_w$ are given in Table 2. The extinction coefficient due to particulate matter, $\lambda_{\mathrm{bio}}$, is dependent on phytoplankton chlorophyll, $P_{\mathrm{chl}}$, and particulate detritus, $R_{\mathrm{C}}^{(2)}$, and is written as

$$\lambda_{\mathrm{bio}} = c_P P_{\mathrm{chl}} + c_{R^{(2)}} R_{\mathrm{C}}^{(2)} , \qquad (4)$$

where $c_P$ and $c_{R^{(2)}}$ are the specific absorption coefficients of phytoplankton chlorophyll and particulate detritus, respectively,
with values given in Table 2.

### 2.1.2 Phytoplankton equations

The phytoplankton LFG in BFM17 is part of the living organic CFF and is composed of separate state variables for the constituents carbon, nitrogen, phosphorous, and chlorophyll, denoted $P_{\mathrm{C}}$, $P_{\mathrm{N}}$, $P_{\mathrm{P}}$, and $P_{\mathrm{chl}}$ respectively (see also Table 1). The governing equations for the constituent state variables are given by:



**Table 3.** List of abbreviations used to indicate physiological and ecological processes in the equations comprising the BFM17 pelagic model.

| Abbreviation | Process |
|---|---|
| gpp | Gross primary production |
| rsp | Respiration |
| prd | Predation |
| rel | Biological release: egestion, excretion, mortality |
| exu | Exudation |
| upt | Uptake |
| lys | Lysis |
| syn | Biochemical synthesis |
| loss | Biochemical loss |
| nit | Nitrification |

1. Phytoplankton functional group in the living organic CFF, carbon constituent (state variable $P_C$):

$$\frac{\partial P_C}{\partial t}\bigg|_{\text{bio}} = \frac{\partial P_C}{\partial t}\bigg|_{CO_2}^{\text{gpp}} - \frac{\partial P_C}{\partial t}\bigg|_{CO_2}^{\text{rsp}} - \frac{\partial P_C}{\partial t}\bigg|_{R_C^{(1)}}^{\text{lys}} - \frac{\partial P_C}{\partial t}\bigg|_{R_C^{(2)}}^{\text{lys}} - \frac{\partial P_C}{\partial t}\bigg|_{R_C^{(1)}}^{\text{exu}} - \frac{\partial P_C}{\partial t}\bigg|_{Z_C}^{\text{prd}},$$ (5)

2. Phytoplankton functional group in the living organic CFF, nitrogen constituent (state variable $P_N$):

$$\frac{\partial P_N}{\partial t}\bigg|_{\text{bio}} = \max\left[0, \frac{\partial P_N}{\partial t}\bigg|_{N^{(2)}}^{\text{upt}} + \frac{\partial P_N}{\partial t}\bigg|_{N^{(3)}}^{\text{upt}}\right] - \frac{\partial P_N}{\partial t}\bigg|_{R_N^{(1)}}^{\text{lys}} - \frac{\partial P_N}{\partial t}\bigg|_{R_N^{(2)}}^{\text{lys}} - \frac{\partial P_N}{\partial t}\bigg|_{Z_N}^{\text{prd}},$$ (6)

3. Phytoplankton functional group in the living organic CFF, phosphorus constituent (state variable $P_P$):

$$\frac{\partial P_P}{\partial t}\bigg|_{\text{bio}} = \max\left[0, \frac{\partial P_P}{\partial t}\bigg|_{N^{(1)}}^{\text{upt}}\right] - \frac{\partial P_P}{\partial t}\bigg|_{R_P^{(1)}}^{\text{lys}} - \frac{\partial P_P}{\partial t}\bigg|_{R_P^{(2)}}^{\text{lys}} - \frac{\partial P_P}{\partial t}\bigg|_{Z_P}^{\text{prd}},$$ (7)

4. Phytoplankton functional group in the living organic CFF, chlorophyll constituent (state variable $P_{chl}$):

$$\frac{\partial P_{chl}}{\partial t}\bigg|_{\text{bio}} = \frac{\partial P_{chl}}{\partial t}\bigg|^{\text{syn}} - \frac{\partial P_{chl}}{\partial t}\bigg|^{\text{loss}},$$ (8)

where the descriptions of each of the source and sink terms are provided in Table 3. The subscript "bio" on the left-hand side terms indicates that these are the total rate expressions associated with all biological processes.

For the evolution of the phytoplankton carbon constituent given by Eq. (5), gross primary production depends on the non-dimensional regulation factors for temperature and light as well as on the maximum photosynthetic growth rate and the phytoplankton carbon instantaneous concentration. This then gives

$$\frac{\partial P_C}{\partial t}\bigg|_{CO_2}^{\text{gpp}} = r_P^{(0)} f_P^{(T)} f_P^{(E)} P_C,$$ (9)





where $r_P^{(0)}$ is the maximum photosynthetic rate for phytoplankton (reported in Table 4) and $f_P^{(T)}$ is the temperature regulation

factor for phytoplankton given by Eq. (1). The term $f_P^{(E)}$ is the light regulation factor for phytoplankton, which is defined following (Jassby and Platt, 1976) as

$$f_P^{(E)} = 1 - \exp\left(-\frac{E_{\text{PAR}}}{E_{\text{K}}}\right), \tag{10}$$

where $E_{\text{PAR}}$ is defined in Eq. (3) and $E_{\text{K}}$ (the "optimal" irradiance) is given by

$$E_{\text{K}} = \left[\frac{r_P^{(0)}}{\alpha_{\text{chl}}^{(0)}}\right]\left(\frac{P_{\text{C}}}{P_{\text{chl}}}\right). \tag{11}$$

The parameter $\alpha_{\text{chl}}^{(0)}$ is the maximum light utilization coefficient and is defined in Table 4.

Phytoplankton respiration is parameterized in Eq. (5) as the sum of the basal respiration and activity respiration rates, namely

$$\left.\frac{\partial P_{\text{C}}}{\partial t}\right|_{\text{CO}_2}^{\text{rsp}} = b_P f_P^{(T)} P_{\text{C}} + \gamma_P \left[\left.\frac{\partial P_{\text{C}}}{\partial t}\right|_{\text{CO}_2}^{\text{gpp}} - \left.\frac{\partial P_{\text{C}}}{\partial t}\right|_{R_{\text{C}}^{(1)}}^{\text{exu}}\right], \tag{12}$$

where $b_P$ is the basal specific respiration rate, $\gamma_P$ is the respired fraction of the gross primary production, the gross primary

production term is given by Eq. (9), and the exudation term is defined below in Eq. (18). Values and descriptions for $b_P$ and $\gamma_P$ are given in Table 4.

Both phytoplankton exudation and lysis, defined below, depend on a multiple nutrient limitation term $f_P^{(\text{N,P})}$. This term allows for the internal storage of nutrients and depends on the respective nutrient limitation terms for both nitrate and phosphate. It is given by $f_P^{(\text{N,P})} = \min\left[f_P^{(\text{N})}, f_P^{(\text{P})}\right]$, where

$$f_P^{(\text{N})} = \min\left\{1, \max\left[0, \frac{P_{\text{N}}/P_{\text{C}} - \phi_{\text{N}}^{(\text{min})}}{\phi_{\text{N}}^{(\text{opt})} - \phi_{\text{N}}^{(\text{min})}}\right]\right\}, \tag{13}$$

$$f_P^{(\text{P})} = \min\left\{1, \max\left[0, \frac{P_{\text{P}}/P_{\text{C}} - \phi_{\text{P}}^{(\text{min})}}{\phi_{\text{P}}^{(\text{opt})} - \phi_{\text{P}}^{(\text{min})}}\right]\right\}. \tag{14}$$

The parameters $\phi_{\text{N}}^{(\text{opt})}$ and $\phi_{\text{P}}^{(\text{opt})}$ are the optimal phytoplankton quotas for nitrogen and phosphorus, respectively, while $\phi_{\text{N}}^{(\text{min})}$ and $\phi_{\text{P}}^{(\text{min})}$ are the minimum possible quotas, below which $f_P^{(\text{N})}$ and $f_P^{(\text{P})}$ are zero. Values for each of these parameters are included in Table 4.

Phytoplankton lysis includes all mortality due to mechanical, viral, and yeast cell disruption processes, and is partitioned between particulate and dissolved detritus. The internal cytoplasm of the cell is released to dissolved detritus, denoted by $R_i^{(1)}$, while structural parts of the cell are released to particulate detritus, denoted by the state variable $R_i^{(2)}$, where $i = \text{C,N,P}$ (see also Table 1). The resulting lysis terms in Eqs. (5)–(7) are then given by

$$\left.\frac{\partial P_i}{\partial t}\right|_{R_i^{(1)}}^{\text{lys}} = \left[1 - \varepsilon_P^{(\text{N,P})}\right]\left[\frac{h_P^{(\text{N,P})}}{f_P^{(\text{N,P})} + h_P^{(\text{N,P})}} d_P^{(0)} P_i\right], \quad i = \text{C,N,P}, \tag{15}$$

$$\left.\frac{\partial P_i}{\partial t}\right|_{R_i^{(2)}}^{\text{lys}} = \varepsilon_P^{(\text{N,P})}\left[\frac{h_P^{(\text{N,P})}}{f_P^{(\text{N,P})} + h_P^{(\text{N,P})}} d_P^{(0)} P_i\right], \quad i = \text{C,N,P}, \tag{16}$$





**Table 4.** Phytoplankton parameters, values, units, and descriptions within the BFM17 pelagic model.

| Symbol | Value | Units | Description |
|--------|-------|-------|-------------|
| $r_P^{(0)}$ | 1.60 | $\text{d}^{-1}$ | Maximum specific photosynthetic rate |
| $b_P$ | 0.05 | $\text{d}^{-1}$ | Basal specific respiration rate |
| $d_P^{(0)}$ | 0.05 | $\text{d}^{-1}$ | Maximum specific nutrient-stress lysis rate |
| $h_P^{(\text{N,P})}$ | 0.10 | - | Nutrient stress threshold |
| $\beta_P$ | 0.05 | - | Excreted fraction of primary production |
| $\gamma_P$ | 0.05 | - | Activity respiration fraction |
| $a_P^{(\text{N})}$ | 0.025 | $\text{m}^3\,(\text{mg C})^{-1}\,\text{d}^{-1}$ | Specific affinity constant for nitrogen |
| $h_P^{(\text{N})}$ | 1.00 | mmol N-NH4 $\text{m}^{-3}$ | Half saturation constant for ammonium uptake |
| $\phi_{\text{N}}^{(\text{min})}$ | $6.87 \times 10^{-3}$ | mmolN $(\text{mg C})^{-1}$ | Minimum nitrogen quota |
| $\phi_{\text{N}}^{(\text{opt})}$ | $1.26 \times 10^{-2}$ | mmolN $(\text{mg C})^{-1}$ | Optimal nitrogen quota |
| $\phi_{\text{N}}^{(\text{max})}$ | $1.0\phi_{\text{N}}^{(\text{opt})}$ | mmolN $(\text{mg C})^{-1}$ | Maximum nitrogen quota |
| $a_P^{(\text{P})}$ | $2.5 \times 10^{-3}$ | $\text{m}^3\,(\text{mg C})^{-1}\,\text{d}^{-1}$ | Specific affinity constant for phosphorus |
| $\phi_{\text{P}}^{(\text{min})}$ | $4.29 \times 10^{-4}$ | mmolP $(\text{mg C})^{-1}$ | Minimum phosphorus quota |
| $\phi_{\text{P}}^{(\text{opt})}$ | $7.86 \times 10^{-4}$ | mmolP $(\text{mg C})^{-1}$ | Optimal phosphorus quota |
| $\phi_{\text{P}}^{(\text{max})}$ | $1.0\phi_{\text{P}}^{(\text{opt})}$ | mmolP $(\text{mg C})^{-1}$ | Maximum phosphorus quota |
| $\alpha_{\text{chl}}^{(0)}$ | $1.52 \times 10^{-5}$ | mgC $(\text{mg chl})^{-1}\,(\mu\text{E})^{-1}\,\text{m}^2$ | Maximum light utilization coefficient |
| $\theta_{\text{chl}}^{(0)}$ | 0.016 | mg chl $(\text{mg C})^{-1}$ | Maximum chlorophyll to carbon quota |

where $h_P^{(\text{N,P})}$ is the nutrient stress threshold and $d_P^{(0)}$ is the maximum specific nutrient-stress lysis rate, both of which are given in Table 4. The term $\varepsilon_P^{(\text{N,P})}$ is a fraction that ensures nutrients within the structural parts of the cell, which are less degradable, are always released as particulate detritus. This fraction is determined by the expression

$$\varepsilon_P^{(\text{N,P})} = \min\left[1, \frac{\phi_{\text{N}}^{(\text{min})}}{P_{\text{N}}/P_{\text{C}}}, \frac{\phi_{\text{P}}^{(\text{min})}}{P_{\text{P}}/P_{\text{C}}}\right], \tag{17}$$

where $\phi_{\text{N}}^{(\text{min})}$ and $\phi_{\text{P}}^{(\text{min})}$ are given in Table 4.

If phytoplankton cannot equilibrate their fixed carbon with sufficient nutrients, this carbon is not assimilated and is instead released in the form of dissolved carbon, denoted by state variable $R_{\text{C}}^{(1)}$, in a process known as exudation. The exudation term in Eq. (5) is parameterized as

$$\left.\frac{\partial P_{\text{C}}}{\partial t}\right|_{R_{\text{C}}^{(1)}}^{\text{exu}} = \left\{\beta_P + (1 - \beta_P)\left[1 - f_P^{(\text{N,P})}\right]\right\}\left.\frac{\partial P_{\text{C}}}{\partial t}\right|_{\text{CO}_2}^{\text{gpp}}, \tag{18}$$

where $\beta_P$ is the excreted fraction of gross primary production, defined in Table 4, and the gross primary production term is again given by Eq. (9).

The nutrient uptake of Eqs. (6) and (7) combines both the intracellular quota (i.e., Droop) and external concentration (i.e., Monod) approaches Baretta-Bekker et al. (1997). The total phytoplankton uptake of nitrogen, represented by the combination





of the two uptake terms in Eq. (6), is the minimum of a diffusion-dependent uptake rate when internal nutrient quotas are low and a rate that is based upon balanced growth needs and any excess uptake, namely

$$
\left.\frac{\partial P_{\mathrm{N}}}{\partial t}\right|_{N^{(2,3)}}^{\mathrm{upt}} = \min\left\{ a_P^{(\mathrm{N})}\left[\frac{h_P^{(\mathrm{N})}}{h_P^{(\mathrm{N})}+N^{(3)}}N^{(2)}+N^{(3)}\right]P_{\mathrm{C}}\,,\, \phi_{\mathrm{N}}^{(\mathrm{max})}G_P + \nu_P\left[\phi_{\mathrm{N}}^{(\mathrm{max})}-\frac{P_{\mathrm{N}}}{P_{\mathrm{C}}}\right]P_{\mathrm{C}}\right\},
\tag{19}
$$

where $a_P^{(\mathrm{N})}$ is the specific affinity for nitrogen, $h_P^{(\mathrm{N})}$ is the half saturation constant for ammonium uptake, and $\phi_{\mathrm{N}}^{(\mathrm{max})}$ is the maximum nitrogen quota; base values for these three parameters are given in Table 4. The net primary productivity $G_P$ in Eq. (19) is given as

$$
G_P = \max\left[0,\, \left.\frac{\partial P_{\mathrm{C}}}{\partial t}\right|_{\mathrm{CO_2}}^{\mathrm{gpp}} - \left.\frac{\partial P_{\mathrm{C}}}{\partial t}\right|_{R_{\mathrm{C}}^{(1)}}^{\mathrm{exu}} - \left.\frac{\partial P_{\mathrm{C}}}{\partial t}\right|_{\mathrm{CO_2}}^{\mathrm{rsp}} - \left.\frac{\partial P_{\mathrm{C}}}{\partial t}\right|_{R_{\mathrm{C}}^{(1)}}^{\mathrm{lys}} - \left.\frac{\partial P_{\mathrm{C}}}{\partial t}\right|_{R_{\mathrm{C}}^{(2)}}^{\mathrm{lys}}\right].
\tag{20}
$$

The specific uptake rate $\nu_P$ appearing in Eq. (19) is given by

$$
\nu_P = f_P^{(\mathrm{T})}r_P^{(0)}.
\tag{21}
$$

It should be noted that only the sum of the two uptake terms, represented by Eq. (19), is required in the governing equation for $P_{\mathrm{N}}$ given by Eq. (6). However, in the governing equations for nitrate and ammonium, denoted $N^{(2)}$ and $N^{(3)}$ (see Table 1) that will be presented later, expressions are required for the individual uptake portions from nitrate and ammonium. When the total phytoplankton nitrogen uptake rate from Eq. (19) is positive, the individual portions from nitrate and ammonium are determined by

$$
\left.\frac{\partial P_{\mathrm{N}}}{\partial t}\right|_{N^{(2)}}^{\mathrm{upt}} = \varepsilon_P\left.\frac{\partial P_{\mathrm{N}}}{\partial t}\right|_{N^{(2,3)}}^{\mathrm{upt}},
\tag{22}
$$

$$
\left.\frac{\partial P_{\mathrm{N}}}{\partial t}\right|_{N^{(3)}}^{\mathrm{upt}} = (1-\varepsilon_P)\left.\frac{\partial P_{\mathrm{N}}}{\partial t}\right|_{N^{(2,3)}}^{\mathrm{upt}},
\tag{23}
$$

where the rates on the right-hand sides are obtained from Eq. (19), and $\varepsilon_P$ is given as

$$
\varepsilon_P = \frac{s_{\mathrm{N}}N^{(2)}}{N^{(3)}+s_{\mathrm{N}}N^{(2)}}.
\tag{24}
$$

The preference for ammonium is defined by the saturation function $s_{\mathrm{N}}$ and is given by

$$
s_{\mathrm{N}} = \frac{h_P^{(\mathrm{N})}}{h_P^{(\mathrm{N})}+N^{(3)}}.
\tag{25}
$$

When the phytoplankton nitrogen uptake rate from Eq. (19) is negative, however, the entire nitrogen uptake goes to the dissolved organic nitrogen pool, $R_{\mathrm{N}}^{(1)}$ [see Eq. (42)].

As with the uptake of nitrogen, phytoplankton uptake of phosphorus in Eq. (7) is the minimum of a diffusion-dependent rate and a balanced growth/excess uptake rate. This uptake comes entirely from one pool and the uptake term in Eq. (7) is correspondingly given by

$$
\left.\frac{\partial P_{\mathrm{P}}}{\partial t}\right|_{N^{(1)}}^{\mathrm{upt}} = \min\left\{ a_P^{(\mathrm{P})}N^{(1)}P_{\mathrm{C}}\,,\, \phi_{\mathrm{P}}^{(\mathrm{max})}G_P + \nu_P\left[\phi_{\mathrm{P}}^{(\mathrm{max})}P_{\mathrm{C}}-P_{\mathrm{P}}\right]\right\},
\tag{26}
$$



where $a_P^{(\mathrm{P})}$ is the specific affinity constant for phosphorous and $\phi_\mathrm{P}^{(\max)}$ is the maximum phosphorous quota. Values for both parameters are given in Table 4. If the uptake rate is negative, the entire phosphorus uptake goes to the dissolved organic phosphorus pool, $R_\mathrm{P}^{(1)}$.

Predation of phytoplankton within BFM17 is solely performed by zooplankton, and each of the predation terms appearing in Eqs. (5)–(7) are equal and opposite to the zooplankton predation terms, namely

$$\left.\frac{\partial P_i}{\partial t}\right|_{Z_i}^{\mathrm{prd}} = -\left.\frac{\partial Z_i}{\partial t}\right|_{P_i}^{\mathrm{prd}}, \quad i = \mathrm{C,N,P}. \tag{27}$$

Equations for the zooplankton predation terms are given in the next section.

Finally, phytoplankton chlorophyll, denoted $P_\mathrm{chl}$ with the rate equation given by Eq. (8), contributes to the definition of the optimal irradiance value in Eq. (11) and of the phytoplankton contribution to the extinction coefficient in Eq. (4). The phytoplankton chlorophyll source term in Eq. (8) is made up of only two terms: chlorophyll synthesis and loss. Net chlorophyll synthesis is a function of acclimation to light n conditions, availability of nutrients, and turnover rate, and is given by

$$\left.\frac{\partial P_\mathrm{chl}}{\partial t}\right|^{\mathrm{syn}} = \rho_\mathrm{chl}\left(1 - \gamma_P\right)\left[\left.\frac{\partial P_\mathrm{C}}{\partial t}\right|_{\mathrm{CO_2}}^{\mathrm{gpp}} - \left.\frac{\partial P_\mathrm{C}}{\partial t}\right|_{R_\mathrm{C}^{(1)}}^{\mathrm{exu}}\right] - \frac{P_\mathrm{chl}}{P_\mathrm{C}}\left[\left.\frac{\partial P_\mathrm{C}}{\partial t}\right|_{R_\mathrm{C}^{(1)}}^{\mathrm{lys}} + \left.\frac{\partial P_\mathrm{C}}{\partial t}\right|_{R_\mathrm{C}^{(2)}}^{\mathrm{lys}} + \left.\frac{\partial P_\mathrm{C}}{\partial t}\right|_{\mathrm{CO_2}}^{\mathrm{rsp}}\right], \tag{28}$$

where $\rho_\mathrm{chl}$ regulates the amount of chlorophyll in the phytoplankton cell and all other terms in the above expression have been defined previously. The term $\rho_\mathrm{chl}$ is computed according to a ratio between the realized photosynthetic rate (i.e., gross primary production) and the maximum potential photosynthesis Geider et al. (1997), and is correspondingly given as

$$\rho_\mathrm{chl} = \theta_\mathrm{chl}^{(0)} \min\left\{1, \frac{(1 - \gamma_P)}{\alpha_\mathrm{chl}^{(0)} E_\mathrm{PAR} P_\mathrm{chl}}\left[\left.\frac{\partial P_\mathrm{C}}{\partial t}\right|_{\mathrm{CO_2}}^{\mathrm{gpp}} - \left.\frac{\partial P_\mathrm{C}}{\partial t}\right|_{R_\mathrm{C}^{(1)}}^{\mathrm{exu}}\right]\right\}, \tag{29}$$

where $\theta_\mathrm{chl}^{(0)}$ is the maximum chlorophyll to carbon quota and $\alpha_\mathrm{chl}^{(0)}$ is the maximum light utilization coefficient, both of which can be found in Table 4. Chlorophyll loss in Eq. (8) is simpler and is just a function of predation, where the amount of chlorophyll transferred back to the infinite sink is proportional to the carbon predated by zooplankton, giving

$$\left.\frac{\partial P_\mathrm{chl}}{\partial t}\right|^{\mathrm{loss}} = \frac{P_\mathrm{chl}}{P_\mathrm{C}}\left.\frac{\partial P_\mathrm{C}}{\partial t}\right|_{Z_\mathrm{C}}^{\mathrm{prd}}. \tag{30}$$

### 2.1.3 Zooplankton equations

The zooplankton LFG group in BFM17 is part of the living organic CFF and is composed of separate state variables for carbon, nitrogen, and phosphorous, denoted $Z_\mathrm{C}$, $Z_\mathrm{N}$, and $Z_\mathrm{P}$, respectively (see also Table 1). The governing equations for the constituent state variables are given by:

5. Zooplankton functional group in the living organic CFF, carbon constituent (state variable $Z_\mathrm{C}$):

$$\left.\frac{\partial Z_\mathrm{C}}{\partial t}\right|_{\mathrm{bio}} = \left.\frac{\partial Z_\mathrm{C}}{\partial t}\right|_{P_\mathrm{C}}^{\mathrm{prd}} - \left.\frac{\partial Z_\mathrm{C}}{\partial t}\right|_{\mathrm{CO_2}}^{\mathrm{rsp}} - \left.\frac{\partial Z_\mathrm{C}}{\partial t}\right|_{R_\mathrm{C}^{(1)}}^{\mathrm{rel}} - \left.\frac{\partial Z_\mathrm{C}}{\partial t}\right|_{R_\mathrm{C}^{(2)}}^{\mathrm{rel}}, \tag{31}$$



6. Zooplankton functional group in the living organic CFF, nitrogen constituent (state variable $Z_\text{N}$):

$$\left.\frac{\partial Z_\text{N}}{\partial t}\right|_\text{bio} = \left.\frac{\partial Z_\text{N}}{\partial t}\right|^\text{prd}_{P_\text{N}} - \left.\frac{\partial Z_\text{N}}{\partial t}\right|^\text{rel}_{R^{(1)}_\text{N}} - \left.\frac{\partial Z_\text{N}}{\partial t}\right|^\text{rel}_{R^{(2)}_\text{N}} - \left.\frac{\partial Z_\text{N}}{\partial t}\right|^\text{rel}_{N^{(3)}} , \qquad (32)$$

7. Zooplankton functional group in the living organic CFF, phosphorus constituent (state variable $Z_\text{P}$):

$$\left.\frac{\partial Z_\text{P}}{\partial t}\right|_\text{bio} = \left.\frac{\partial Z_\text{P}}{\partial t}\right|^\text{prd}_{P_\text{P}} - \left.\frac{\partial Z_\text{P}}{\partial t}\right|^\text{rel}_{R^{(1)}_\text{P}} - \left.\frac{\partial Z_\text{P}}{\partial t}\right|^\text{rel}_{R^{(2)}_\text{P}} - \left.\frac{\partial Z_\text{P}}{\partial t}\right|^\text{rel}_{N^{(1)}} , \qquad (33)$$

where, once more, descriptions of each of the source and sink terms are provided in Table 3.

Zooplankton predation of phytoplankton, which appears as the first term in each of Eqs. (31)–(33), primarily depends on the availability of phytoplankton and their capture efficiency, and is expressed as

$$\left.\frac{\partial Z_i}{\partial t}\right|^\text{prd}_{P_i} = \frac{P_i}{P_\text{C}} \left[ f^{(T)}_Z r^{(0)}_Z \delta_{Z,\text{P}} \frac{P_\text{C}}{P_\text{C} + h^{(F)}_Z} Z_\text{C} \right] , \quad i = \text{C}, \text{N}, \text{P}, \qquad (34)$$

where $r^{(0)}_Z$ is the potential specific growth rate and $h^{(F)}_Z$ is the Michaelis constant for total food ingestion. These parameters

and their base values are included in Table 5. Here, $f^{(T)}_Z$ is the temperature regulating factor for zooplankton growth given by Eq. (1). The total food availability can be expressed as $\delta_{Z,P} P_\text{C}$, where $\delta_{Z,P}$ is the prey availability of phytoplankton and is included in Table 5.

Zooplankton respiration is the sum of active and basal metabolism rates, where active respiration is the cost of nutrient ingestion, or predation. The resulting respiration rate is given by

$$\left.\frac{\partial Z_\text{C}}{\partial t}\right|^\text{rsp}_{\text{CO}_2} = (1 - \eta_Z - \beta_Z) \left.\frac{\partial Z_\text{C}}{\partial t}\right|^\text{prd}_{P_c} + b_Z f^{(T)}_Z Z_\text{C} , \qquad (35)$$

where $\eta_Z$ is the assimilation efficiency, $\beta_Z$ is the excreted fraction uptake, and $b_Z$ is the basal specific respiration rate. All three parameters are included in Table 5.

The biological release terms in Eqs. (31)–(33) are the sum of zooplankton excretion, egestion, and mortality. Excretion and egestion are the portions of ingested nutrients, resulting from predation, that have not been assimilated or used for respiration.

Zooplankton mortality is parameterized as the sum of a constant mortality rate and an oxygen-dependent regulation factor given by

$$f^{(\text{O})}_Z = \frac{O}{O + h_O} , \qquad (36)$$

where $O$ represents the oxygen constituent of the dissolved gas in the inorganic CFF and $h_O$ is the half saturation coefficient for chemical processes given in Table 6. The total biological release is then partitioned into particulate and dissolved organic

matter, giving

$$\left.\frac{\partial Z_i}{\partial t}\right|^\text{rel}_{R^{(1)}_i} = \varepsilon^{(i)}_Z \left\{ \beta_Z \left.\frac{\partial Z_i}{\partial t}\right|^\text{prd}_{P_i} + d_Z + d^{(0)}_Z \left[ 1 - f^{(\text{O})}_Z \right] f^{(T)}_Z Z_i \right\} , \quad i = \text{C}, \text{N}, \text{P}, \qquad (37)$$

$$\left.\frac{\partial Z_i}{\partial t}\right|^\text{rel}_{R^{(2)}_i} = \left[ 1 - \varepsilon^{(i)}_Z \right] \left.\frac{\partial Z_i}{\partial t}\right|^\text{rel}_{R^{(1)}_i} , \quad i = \text{C}, \text{N}, \text{P}, \qquad (38)$$





**Table 5.** Zooplankton parameters, values, units, and descriptions within the BFM17 pelagic model.

| Symbol | Value | Unit | Description |
|--------|-------|------|-------------|
| $b_Z$ | 0.02 | d$^{-1}$ | Basal specific respiration rate |
| $r_Z^{(0)}$ | 2.00 | d$^{-1}$ | Potential specific growth rate |
| $d_Z^{(0)}$ | 0.25 | d$^{-1}$ | Oxygen dependent specific mortality rate |
| $d_Z$ | 0.05 | d$^{-1}$ | Specific mortality rate |
| $\eta_Z$ | 0.50 | - | Assimilation efficiency |
| $\beta_Z$ | 0.25 | - | Fraction of activity excretion |
| $\varepsilon_Z^{\mathrm{C}}$ | 0.60 | - | Partition between dissolved and particulate excretion of C |
| $\varepsilon_Z^{\mathrm{N}}$ | 0.72 | - | Partition between dissolved and particulate excretion of N |
| $\varepsilon_Z^{\mathrm{P}}$ | 0.832 | - | Partition between dissolved and particulate excretion of P |
| $h_Z^{(F)}$ | 200.0 | mg C m$^{-3}$ | Michaelis constant for total food ingestion |
| $\delta_{Z,P}$ | 1.00 | - | Availability of phytoplankton to zooplankton |
| $\nu_Z^{(\mathrm{P})}$ | 1.0 | d$^{-1}$ | Specific rate constant for phosphorous excretion |
| $\nu_Z^{(\mathrm{N})}$ | 1.0 | d$^{-1}$ | Specific rate constant for nitrogen excretion |
| $\varphi_{\mathrm{P}}^{(\mathrm{opt})}$ | $7.86 \times 10^{-4}$ | mmolP (mg C)$^{-1}$ | Optimal phosphorous quota |
| $\varphi_{\mathrm{N}}^{(\mathrm{opt})}$ | 0.0126 | mmolN (mg C)$^{-1}$ | Optimal nitrogen quota |

where $\varepsilon_Z^{(i)}$ is the fraction excreted to the dissolved pool, $d_Z$ is the specific mortality rate, and $d_Z^{(0)}$ is the oxygen dependent specific morality rate. Base values for each parameter are given in Table 5.

The zooplankton also excrete into the nutrient pools of phosphate, $N^{(1)}$, and ammonium, $N^{(3)}$. These effects are represented by the final terms of Eqs. (32) and (33), which are parameterized by

$$\frac{\partial Z_{\mathrm{N}}}{\partial t}\bigg|_{N^{(3)}}^{\mathrm{rel}} = \nu_Z^{(\mathrm{N})}\max\left[0, \frac{Z_{\mathrm{N}}}{Z_{\mathrm{C}}} - \varphi_{\mathrm{N}}^{(\mathrm{opt})}\right] Z_{\mathrm{N}}, \tag{39}$$

$$\frac{\partial Z_{\mathrm{P}}}{\partial t}\bigg|_{N^{(1)}}^{\mathrm{rel}} = \nu_Z^{(\mathrm{P})}\max\left[0, \frac{Z_{\mathrm{P}}}{Z_{\mathrm{C}}} - \varphi_{\mathrm{P}}^{(\mathrm{opt})}\right] Z_{\mathrm{P}}, \tag{40}$$

where $\nu_Z^{(\mathrm{N})}$ and $\nu_Z^{(\mathrm{P})}$ are specific rate constants and $\varphi_{\mathrm{N}}^{(\mathrm{opt})}$ and $\varphi_{\mathrm{P}}^{(\mathrm{opt})}$ are the optimal zooplankton quotas for nitrogen and
phosphorous, respectively. All four parameters are included in Table 5.

### 2.1.4   Dissolved organic matter equations

The governing equations for the three constituents of dissolved organic matter are given by:

8. Dissolved matter in non-living organic CFF, carbon constituent [state variable $R_{\mathrm{C}}^{(1)}$]:

$$\frac{\partial R_{\mathrm{C}}^{(1)}}{\partial t}\bigg|_{\mathrm{bio}} = \frac{\partial P_{\mathrm{C}}}{\partial t}\bigg|_{R_{\mathrm{C}}^{(1)}}^{\mathrm{lys}} + \frac{\partial P_{\mathrm{C}}}{\partial t}\bigg|_{R_{\mathrm{C}}^{(1)}}^{\mathrm{exu}} + \frac{\partial Z_{\mathrm{C}}}{\partial t}\bigg|_{R_{\mathrm{C}}^{(1)}}^{\mathrm{rel}} - \alpha_{R^{(1)}}^{(\mathrm{sink_C})} R_{\mathrm{C}}^{(1)}, \tag{41}$$





**Table 6.** Values, units, and descriptions for dissolved organic matter, particulate organic matter, and nutrient parameters within the BFM17 pelagic model.

| Symbol | Value | Units | Description |
|---|---|---|---|
| $\alpha_{R^{(1)}}^{(\text{sink}_C)}$ | 0.05 | $\text{d}^{-1}$ | Specific remineralization rate of dissolved carbon |
| $\zeta_{N^{(1)}}$ | 0.05 | $\text{d}^{-1}$ | Specific remineralization rate of dissolved phosphorus |
| $\zeta_{N^{(3)}}$ | 0.05 | $\text{d}^{-1}$ | Specific remineralization rate of dissolved nitrogen |
| $\alpha_{R^{(2)}}^{(\text{sink}_C)}$ | 0.1 | $\text{d}^{-1}$ | Specific remineralization rate of particulate carbon |
| $\xi_{N^{(1)}}$ | 0.1 | $\text{d}^{-1}$ | Specific remineralization rate of particulate phosphorus |
| $\xi_{N^{(3)}}$ | 0.1 | $\text{d}^{-1}$ | Specific remineralization rate of particulate nitrogen |
| $\Lambda_{N^{(3)}}^{(\text{nit})}$ | 0.01 | $\text{d}^{-1}$ | Specific nitrification rate at 10 °C |
| $h_O$ | 10.0 | $\text{mmolO}_2 \text{ m}^{-3}$ | Half saturation for chemical processes |
| $\Omega_{\text{C}}^{(O)}$ | 12.0 | $\text{mmolO}_2 \text{ mgC}^{-1}$ | Stoichiometric coefficient for oxygen reaction |
| $\Omega_{\text{N}}^{(O)}$ | 2.0 | $\text{mmolO}_2 \text{ mmolN}^{-1}$ | Stoichiometric coefficient for nitrification reaction |

9. Dissolved matter in non-living organic CFF, nitrogen constituent [state variable $R_{\text{N}}^{(1)}$]:

$$\left.\frac{\partial R_{\text{N}}^{(1)}}{\partial t}\right|_{\text{bio}} = \left.\frac{\partial P_{\text{N}}}{\partial t}\right|_{R_{\text{N}}^{(1)}}^{\text{lys}} + \left.\frac{\partial Z_{\text{N}}}{\partial t}\right|_{R_{\text{N}}^{(1)}}^{\text{rel}} - \min\left[0, \left.\frac{\partial P_{\text{N}}}{\partial t}\right|_{N^{(2)}}^{\text{upt}} + \left.\frac{\partial P_{\text{N}}}{\partial t}\right|_{N^{(3)}}^{\text{upt}}\right] - \zeta_{N^{(3)}} R_{\text{N}}^{(1)}, \tag{42}$$

10. Dissolved matter in non-living organic CFF, phosphorus constituent [state variable $R_{\text{P}}^{(1)}$]:

$$\left.\frac{\partial R_{\text{P}}^{(1)}}{\partial t}\right|_{\text{bio}} = \left.\frac{\partial P_{\text{P}}}{\partial t}\right|_{R_{\text{P}}^{(1)}}^{\text{lys}} + \left.\frac{\partial Z_{\text{P}}}{\partial t}\right|_{R_{\text{P}}^{(1)}}^{\text{rel}} - \min\left[0, \left.\frac{\partial P_{\text{P}}}{\partial t}\right|_{N^{(1)}}^{\text{upt}}\right] - \zeta_{N^{(1)}} R_{\text{P}}^{(1)}. \tag{43}$$

All terms except for the last terms in each of these equations, representing remineralization, have been defined in previous
sections. Remineralization of dissolved organic matter by bacteria is parameterized within BFM17 as a rate that is proportional to the local concentration of that dissolved constituent. In Eq. (41), remineralization is parameterized as $\alpha_{R^{(1)}}^{(\text{sink}_C)} R_{\text{C}}^{(1)}$, where $\alpha_{R^{(1)}}^{(\text{sink}_C)}$ is a constant that controls the rate at which dissolved carbon is remineralized and returned to the pool of carbon; this constant is given in Table 6. In Eqs. (42) and (43), remineralization is represented by the parameters $\zeta_{N^{(3)}}$ and $\zeta_{N^{(1)}}$, which are the specific remineralization rates of dissolved ammonium and phosphate, respectively. These rates are also included in Table 6

**2.1.5   Particulate organic matter equations**

The governing equations for the three constituents of particulate organic matter are given by:

11. Particulate matter in non-living organic CFF, carbon constituent [state variable $R_{\text{C}}^{(2)}$]:

$$\left.\frac{\partial R_{\text{C}}^{(2)}}{\partial t}\right|_{\text{bio}} = \left.\frac{\partial P_{\text{C}}}{\partial t}\right|_{R_{\text{C}}^{(2)}}^{\text{lys}} + \left.\frac{\partial Z_{\text{C}}}{\partial t}\right|_{R_{\text{C}}^{(2)}}^{\text{rel}} - \alpha_{R^{(2)}}^{(\text{sink}_C)} R_{\text{C}}^{(2)}, \tag{44}$$





12. Particulate matter in non-living organic CFF, nitrogen constituent [state variable $R_{\mathrm{N}}^{(2)}$]:

$$\left.\frac{\partial R_{\mathrm{N}}^{(2)}}{\partial t}\right|_{\mathrm{bio}} = \left.\frac{\partial P_{\mathrm{N}}}{\partial t}\right|_{R_{\mathrm{N}}^{(2)}}^{\mathrm{lys}} + \left.\frac{\partial Z_{\mathrm{N}}}{\partial t}\right|_{R_{\mathrm{N}}^{(2)}}^{\mathrm{rel}} - \xi_{N^{(3)}} R_{\mathrm{N}}^{(2)}, \tag{45}$$

13. Particulate matter in non-living organic CFF, phosphorus constituent [state variable $R_{\mathrm{P}}^{(2)}$]:

$$\left.\frac{\partial R_{\mathrm{P}}^{(2)}}{\partial t}\right|_{\mathrm{bio}} = \left.\frac{\partial P_{\mathrm{P}}}{\partial t}\right|_{R_{\mathrm{P}}^{(2)}}^{\mathrm{lys}} + \left.\frac{\partial Z_{\mathrm{P}}}{\partial t}\right|_{R_{\mathrm{P}}^{(2)}}^{\mathrm{rel}} - \xi_{N^{(1)}} R_{\mathrm{P}}^{(2)}. \tag{46}$$

Once again, all terms except for the final remineralization terms in each equation have been defined in previous sections. Remineralization of particular organic matter by bacteria is parameterized within BFM17 as a rate that is proportional to the local concentration of that particulate constituent. In Eq. (44), remineralization is parameterized by $\alpha_{R^{(2)}}^{(\mathrm{sink}_{\mathrm{C}})} R_{\mathrm{C}}^{(2)}$, where $\alpha_{R^{(2)}}^{(\mathrm{sink}_{\mathrm{C}})}$ is a constant that controls the rate at which the particulate carbon is remineralized. The base value for this constant is provided in Table 6. The parameters $\xi_{N^{(3)}}$ and $\xi_{N^{(1)}}$ are the specific remineralization rates of particulate ammonium and phosphate, respectively. The specific remineralization rates for particulate organic matter are also presented in Table 6.

### 2.1.6 Dissolved gas and nutrient equations

The only dissolved gas resolved by BFM17 is oxygen, $O$, (carbon dioxide is treated as an infinite source/sink) and the dissolved nutrients in the model are phosphate, $N^{(1)}$, nitrate, $N^{(2)}$, and ammonium, $N^{(3)}$ (see also Table 1). Governing equations for each of these state variables are given by:

14. Dissolved gas in the inorganic CFF, oxygen constituent (state variable $O$):

$$\left.\frac{\partial O}{\partial t}\right|_{\mathrm{bio}} = \left.\frac{\partial O}{\partial t}\right|^{\mathrm{wind}} + \Omega_{\mathrm{C}}^{(O)} \left[ \left.\frac{\partial P_{\mathrm{C}}}{\partial t}\right|_{\mathrm{CO_2}}^{\mathrm{gpp}} - \left.\frac{\partial P_{\mathrm{C}}}{\partial t}\right|_{\mathrm{CO_2}}^{\mathrm{rsp}} - \left.\frac{\partial Z_{\mathrm{C}}}{\partial t}\right|_{\mathrm{CO_2}}^{\mathrm{rsp}} - \alpha_{R^{(2)}}^{(\mathrm{sink}_{\mathrm{C}})} R_{\mathrm{C}}^{(2)} - \alpha_{R^{(1)}}^{(\mathrm{sink}_{\mathrm{C}})} R_{\mathrm{C}}^{(1)} \right]$$
$$- \Omega_{\mathrm{N}}^{(O)} \left.\frac{\partial N^{(3)}}{\partial t}\right|_{N^{(2)}}^{\mathrm{nit}}, \tag{47}$$

15. Dissolved nutrient in the inorganic CFF, phosphate constituent (state variable $N^{(1)}$):

$$\left.\frac{\partial N^{(1)}}{\partial t}\right|_{\mathrm{bio}} = - \left.\frac{\partial P_{\mathrm{P}}}{\partial t}\right|_{N^{(1)}}^{\mathrm{upt}} + \zeta_{N^{(1)}} R_{\mathrm{P}}^{(1)} + \xi_{N^{(1)}} R_{\mathrm{P}}^{(2)} + \left.\frac{\partial Z_{\mathrm{P}}}{\partial t}\right|_{N^{(1)}}^{\mathrm{rel}}, \tag{48}$$

16. Dissolved nutrient in the inorganic CFF, nitrate constituent (state variable $N^{(2)}$):

$$\left.\frac{\partial N^{(2)}}{\partial t}\right|_{\mathrm{bio}} = - \left.\frac{\partial P_{\mathrm{N}}}{\partial t}\right|_{N^{(2)}}^{\mathrm{upt}} + \left.\frac{\partial N^{(2)}}{\partial t}\right|_{N^{(3)}}^{\mathrm{nit}}, \tag{49}$$

17. Dissolved nutrient in the inorganic CFF, ammonium constituent (state variable $N^{(3)}$):

$$\left.\frac{\partial N^{(3)}}{\partial t}\right|_{\mathrm{bio}} = - \left.\frac{\partial P_{\mathrm{N}}}{\partial t}\right|_{N^{(3)}}^{\mathrm{upt}} + \zeta_{N^{(3)}} R_{\mathrm{N}}^{(1)} + \xi_{N^{(3)}} R_{\mathrm{N}}^{(2)} + \left.\frac{\partial Z_{\mathrm{N}}}{\partial t}\right|_{N^{(3)}}^{\mathrm{rel}} - \left.\frac{\partial N^{(3)}}{\partial t}\right|_{N^{(2)}}^{\mathrm{nit}}. \tag{50}$$





Aeration of the surface layer by wind, $\partial O/\partial t|^{\text{wind}}$, is parameterized as described in Refs. (Wanninkhof, 1992, 2014). In a zero-dimensional model it is a source term for dissolved oxygen and so belongs in Eq. (47). However, in any model of one dimension or more it should be treated as a surface boundary condition for dissolved oxygen and so belongs in Eq. (69) and

should be omitted from Eq. (47). The parameters $\Omega_C^{(O)}$ and $\Omega_N^{(O)}$ are stoichiometric coefficients used to convert units of carbon to units of oxygen and nitrogen, respectively. All terms in the above equations have been defined in previous sections, except for nitrification. Nitrification is a source term for nitrate and is parameterized as a sink of ammonium and oxygen as

$$\left.\frac{\partial N^{(2)}}{\partial t}\right|_{N^{(3)}}^{\text{nit}} = \left.\frac{\partial N^{(3)}}{\partial t}\right|_{N^{(2)}}^{\text{nit}} = \Lambda_{N^{(3)}}^{\text{(nit)}} f_N^{(T)} f_Z^{(O)} N^{(3)}, \tag{51}$$

where $\Lambda_{N^{(3)}}^{\text{(nit)}}$ is the specific nitrification rate, given in Table 6. The terms $f_N^{(T)}$ and $f_Z^{(O)}$ are defined in Eqs. (2) and (36),

respectively.

## 2.2 Zero-Dimensional Test of the BFM17

As an initial test of BFM17, the model was integrated in a 0D (i.e., time only) test for 10 years using sinusoidal forcing for the temperature (in units of °C), salinity (psu), 10 m wind-speed (m s$^{-1}$), and PAR (W m$^{-2}$) cycles. This forcing is implemented as

$$F^{(\text{var})}(t) = \left[F_s^{(\text{var})} + F_w^{(\text{var})}\right] - 0.5\left[F_s^{(\text{var})} - F_w^{(\text{var})}\right]\cos\left(tR\right), \tag{52}$$

where $F^{(\text{var})}$ is the annually varying forcing term, 'var' indicates the variable of interest, corresponding to temperature ('temp'), salinity ('sal'), wind speed ('wind'), and PAR. In Eq. (52), $F_w^{(\text{var})}$ and $F_s^{(\text{var})}$ are, respectively, the winter and summer extreme values for the forcing term considered, $0 \le t \le 360$ is the time, and $R = \pi/180$. The winter and summer values were chosen to be similar to those found in the observational data described later in Section 4, with $[F_w^{(\text{temp})}, F_s^{(\text{temp})}] = [10°C, 30°C]$,

$[F_w^{(\text{sal})}, F_s^{(\text{sal})}] = [37\text{ psu}, 36.5\text{ psu}]$, $[F_w^{(\text{wind})}, F_s^{(\text{wind})}] = [6\text{ m s}^{-1}, 2\text{ m s}^{-1}]$, and $[F_w^{(\text{PAR})}, F_s^{(\text{PAR})}] = [10\text{ W m}^{-2}, 120\text{ W m}^{-2}]$. Note that, in the 0D framework, the wind forcing does not constrain the biogeochemical dynamics, but does play a role in oxygen exchange with the atmosphere, defined according to Wanninkhof (Wanninkhof, 1992, 2014).

Initial values for chlorophyll, oxygen, phosphate, and nitrate were taken to be similar to values from the observational data, with $P_{\text{chl}} = 0.2$ mg Chl-$a$ m$^{-3}$, $O = 230$ mmol O$_2$ m$^{-3}$, $N^{(1)} = 0.06$ mmol P m$^{-3}$, and $N^{(2)} = 1.0$ mmol N m$^{-3}$). Phytoplankton

carbon was calculated using the maximum chlorophyll to carbon ratio, $\theta_{\text{chl}}^{(0)}$ in Table 4. Initial values for zooplankton carbon, dissolved carbon, and particulate organic carbon were assumed to be the same as the phytoplankton carbon. Ammonium was assumed to have the same initial concentration as phosphate. All other constituents were calculated using their respective optimal ratios in Tables 4 and 5.

Figure 2 shows the seasonal cycle of surface chlorophyll, zooplankton carbon, and nitrate over the last 4 years of the 10-year

simulation period, indicating that a self-consistent and stable seasonal cycle with reasonable ecosystem values can be attained by the reduced model. Figures 2(a) and (c) also show monthly averaged values taken from the observational data described in Section 4. Although the agreement between the 0D BFM17 model and the observations is not perfect, both are qualitatively similar and close in magnitude, providing confidence in the accuracy of the model despite the lower fidelity of the 0D test.



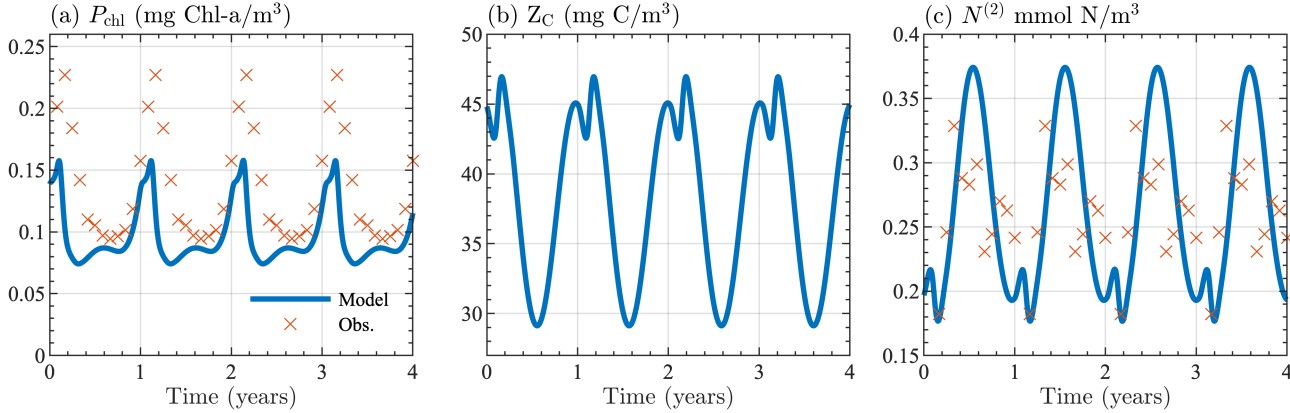

**Figure 2.** Seasonal cycle of surface (a) chlorophyll, (b) zooplankton carbon, and (c) nitrate from the 0D test of BFM17. Results are shown for the last 4 years of the 10-year simulation. Panels (a) and (c) show monthly averaged values taken from the observational data described in Section 4.

## 3 Coupled Physical-Biogeochemical Flux Model

In the following sections, we describe the coupled 1D physical and biogeochemical model used for the comparisons with the observational data. The coupled physical and biogeochemical model is a time-depth model that integrates in time the generic equation for all biological state variables, denoted $A_j$, given by

$$\frac{\partial A_j}{\partial t} = \frac{\partial A_j}{\partial t}\bigg|_{\text{bio}} - \left[W + W_E + v^{(\text{set})}\right]\frac{\partial A_j}{\partial z} + \frac{\partial}{\partial z}\left(K_H \frac{\partial A_j}{\partial z}\right), \tag{53}$$

where the first term on the right-hand side accounts for sources and sinks within each species due to biological and chemical
reactions. Although the BFM17 formulation and model results are the primary focus of the present study, we also perform coupled physical-biogeochemical simulations using BFM56 for comparison. Equation (53) applies to all 17 state variables in BFM17, as well as to all 56 state variables in BFM56. Consequently, the only differences between the biophysical models with BFM17 and BFM56 are the number of state variables being tracked and the equations used to calculate the biological forcing terms. The specific forms of Eq. (53) for each of the 17 species in BFM17 are discussed in Section 2.1, and the specific forms
of this equation for each of the 56 species in BFM56 were previously discussed in Vichi et al. (2007). The parameters used in BFM56 correspond to the values provided in Tables 4–6, with the remaining undefined parameters (since BFM56 includes many more model parameters than BFM17) based on values from Mussap et al. (2016).

The vertical velocities $W$ and $W_E$ in Eq. (53) are the large-scale general circulation and mesoscale eddy vertical velocities, respectively. The range of values for each of these velocities are included in Table 7 and the corresponding depth profiles are discussed in Section 4.3. The settling velocity, $v^{(\text{set})}$, in Eq. (53) is only non-zero for the three constituents of particulate
organic matter, namely, $R_{\text{C,N,P}}^{(2)}$, and its value is given in Table 7. We assume $v^{(\text{set})} = 0$ for zooplankton, since zooplankton



**Table 7.** Values, units, and descriptions for parameters used in the combined physical–BFM17 model.

| Symbol | Value | Units | Description |
|---|---|---|---|
| $v^{(\text{set})}$ | -1.00 | m d$^{-1}$ | Settling velocity of particulate detritus |
| $W$ | $-0.02 - 0$ | m d$^{-1}$ | Imposed general circulation vertical velocity |
| $W_E$ | $0 - 0.1$ | m d$^{-1}$ | Imposed mesoscale circulation vertical velocity |
| $\lambda_O$ | 0.06 | m d$^{-1}$ | Relaxation constant for oxygen at bottom |
| $\lambda_{N^{(1)}}$ | 0.06 | m d$^{-1}$ | Relaxation constant for phosphate at bottom |
| $\lambda_{N^{(2)}}$ | 0.06 | m d$^{-1}$ | Relaxation constant for nitrate at bottom |
| $\kappa_{N^{(3)}}$ | 0.05 | m$^2$ s$^{-1}$ | Relaxation diffusivity for ammonium at bottom |

actively swim and oppose their own sinking velocity. Finally, $K_H$ in Eq. (53) is the vertical eddy diffusivity term calculated by the model, and is described in more detail later in this section.

To obtain the complete 1D biophysical model, BFM17 has been coupled with a modification of the three-dimensional Princeton Ocean Model (POM) (Blumberg and Mellor, 1987) that considers only the vertical and time dimensions; that is, the evolution of the system in the $(z, t)$ space. It is well known that the primary calibration dimension in marine ocean biogeochemistry is along the vertical direction, as shown in several previous calibration and validation exercises (Vichi et al., 2003; Triantafyllou et al., 2003; Mussap et al., 2016).

The 1D POM solver (POM-1D) is used to calculate the vertical structure of the two horizontal velocity components, denoted $U$ and $V$, the potential temperature, $T$, salinity, $S$, density, $\rho$, turbulent kinetic energy, $q^2/2$, and mixing length scale, $\ell$. In this model adaptation, vertical temperature and salinity profiles are imposed from given climatological monthly profiles, as previously done in Mussap et al. (2016) and Bianchi et al. (2005). The model computes only the time evolution of the horizontal velocity components, the turbulent kinetic energy and the mixing length scale, all of which are used to compute the turbulent diffusivity term, $K_H$, required in Eq. (53). In this configuration, POM-1D is called "diagnostic" since temperature and salinity are prescribed. Furthermore, pressure effects are neglected in the density equation and the buoyancy gradients and temperature are used in place of potential temperature since we consider only the upper water column.

In diagnostic mode, POM-1D solves the momentum equations for $U$ and $V$ given by

$$\frac{\partial U}{\partial t} - fV = \frac{\partial}{\partial z}\left(K_M \frac{\partial U}{\partial z}\right), \tag{54}$$

$$\frac{\partial V}{\partial t} + fU = \frac{\partial}{\partial z}\left(K_M \frac{\partial V}{\partial z}\right), \tag{55}$$

where $f = 2\Omega \sin\phi$ is the Coriolis force, $\Omega$ is the angular velocity of the Earth, and $\phi$ is the latitude. The vertical viscosity $K_M$ and diffusivity $K_H$ are calculated using the closure hypothesis of Mellor and Yamada (1982) as

$$K_M = q\,l\,S_M, \tag{56}$$

$$K_H = q\,l\,S_H, \tag{57}$$





where $q$ is the turbulent velocity and $S_H$ and $S_M$ are stability functions written as

$$S_M \left[ 1 - 9A_1 A_2 G_H \right] - S_H \left[ (18A_1^2 + 9A_1 A_2) G_H \right] = A_1 \left[ 1 - 3C_1 - 6A_1/B_1 \right], \tag{58}$$

$$S_H \left[ 1 - (3A_2 B_2 + 18A_1 A_2) G_H \right] = A_2 \left[ 1 - 6A_1/B_1 \right]. \tag{59}$$

The coefficients in the above expressions are $(A_1, B_1, A_2, B_2, C_1) = (0.92, 16.6, 0.74, 10.1, 0.08)$, with

$$G_H = \frac{l^2}{q^2} \frac{g}{\rho_0} \frac{\partial \rho}{\partial z}, \tag{60}$$

where $\rho_0 = 1025$ kg m$^{-3}$, $g = 9.81$ m s$^{-2}$. Following Mellor (2001), $G_H$ is limited to have a maximum value of 0.028. The equation of state relating $\rho$ to $T$ and $S$ is nonlinear (Mellor, 1991) and given by

$$\begin{aligned}
\rho =~ & 999.8 + (6.8 \times 10^{-2} - 9.1 \times 10^{-3} T + 1.0 \times 10^{-4} T^2 - 1.1 \times 10^{-6} T^3 + 6.5 \times 10^{-9} T^4) T \\
& + (0.8 - 4.1 \times 10^{-3} T + 7.6 \times 10^{-5} T^2 - 8.3 \times 10^{-7} T^3 + 5.4 \times 10^{-9} T^4) S \\
& + (-5.7 \times 10^{-3} + 1.0 \times 10^{-4} T - 1.6 \times 10^{-6} T^2) S^{1.5} + 4.8 \times 10^{-4} S^2,
\end{aligned} \tag{61}$$

where the polynomial constants have been written only up to the first digit. For a more precise reproduction of these constants, the reader is referred to Mellor (1991). Finally, the governing equations solved to obtain the turbulence variables $q^2/2$ and $\ell$ are

$$\frac{\partial}{\partial t} \left( \frac{q^2}{2} \right) = \frac{\partial}{\partial z} \left[ K_q \frac{\partial}{\partial z} \left( \frac{q^2}{2} \right) \right] + K_M \left[ \left( \frac{\partial U}{\partial z} \right)^2 + \left( \frac{\partial V}{\partial z} \right)^2 \right] + \frac{g}{\rho_0} K_H \frac{\partial \rho}{\partial z} - \frac{q^3}{B_1 \ell}, \tag{62}$$

$$\frac{\partial}{\partial t} \left( q^2 \ell \right) = \frac{\partial}{\partial z} \left[ K_q \frac{\partial}{\partial z} \left( q^2 \ell \right) \right] + E_1 \ell K_M \left[ \left( \frac{\partial U}{\partial z} \right)^2 + \left( \frac{\partial V}{\partial z} \right)^2 \right] + E_1 \ell \frac{g}{\rho_0} K_H \frac{\partial \rho}{\partial z} - \frac{q^3}{B_1} \widetilde{W}, \tag{63}$$

where $K_q = \kappa K_H$ is the vertical diffusivity for turbulence variables, $\kappa = 0.4$ is the von Karman constant, and $\widetilde{W} = \left[ 1 + E_2 \ell^2 / \kappa^2 \left( 1/|z| + 1/|z - H| \right)^2 \right]$ with $(E_1, E_2) = (1.8, 1.33)$. In Eqs. (62) and (63), the time rate of change of the turbulence quantities is equal to the diffusion of turbulence (the first term on the right hand side of both equations), the shear and buoyancy turbulence production (second and third terms), and the dissipation (the fourth term). This is a second-order turbulence closure model that was formulated by Mellor (2001) as a particular case of the Mellor and Yamada (1982) model for upper ocean mixing.

Boundary conditions for the horizontal velocities $\boldsymbol{U} = (U, V)$ and the turbulence quantities are

$$K_M \frac{\partial \boldsymbol{U}}{\partial z} \bigg|_{z=0} = \boldsymbol{\tau}_w, \tag{64}$$

$$K_M \frac{\partial \boldsymbol{U}}{\partial z} \bigg|_{z=z_{\text{end}}} = 0, \tag{65}$$

$$\left( q^2, q^2 \ell \right) \big|_{z=0} = \left( B_1^{2/3} \frac{|\boldsymbol{\tau}_w|}{C_d}, 0 \right), \tag{66}$$

$$\left( q^2, q^2 \ell \right) \big|_{z=z_{\text{end}}} = 0, \tag{67}$$

where $\boldsymbol{\tau}_w = C_d |\boldsymbol{u}_w| \boldsymbol{u}_w$ is the surface wind stress, $\boldsymbol{u}_w$ is the surface wind vector, $C_d$ is a constant drag coefficient chosen to be $2.5 \times 10^{-3}$, and $z = 0$ and $z = z_{\text{end}}$ denote the locations of the surface and the greatest depth modeled, respectively.





For all variables except oxygen, surface boundary conditions for the coupled model variable $A_j$ are

$$K_H \frac{\partial A_j}{\partial z}\bigg|_{z=0} = 0. \tag{68}$$

By contrast, the surface boundary condition for oxygen has the form

$$K_H \frac{\partial O}{\partial z}\bigg|_{z=0} = \Phi_O, \tag{69}$$

where $\Phi_O$ is the air-sea interface flux of oxygen computed according to Wanninkhof (1992, 2014). The bottom (i.e., greatest depth) boundary conditions for phytoplankton, zooplankton, dissolved organic matter, and particulate organic matter are

$$K_H \frac{\partial A_j}{\partial z}\bigg|_{z=z_{\text{end}}} = 0. \tag{70}$$

This boundary condition was chosen since it allows removal of the scalar quantity $A_j$ through the bottom boundary of the

domain. This can be seen by integrating Eq. (53) over the boundary layer depth using the boundary condition above, giving

$$\frac{\partial}{\partial t} \int_{z=z_{\text{end}}}^{z=0} A_j dz = \left[W + W_E + v^{(\text{set})}\right] A_j\big|_{z=z_{\text{end}}}, \tag{71}$$

where the biological part of Eq. (53) has been neglected and the resulting temporal change in the integrated scalar $A_j$ is negative since $|(W + W_E)| < |v^{(\text{set})}|$, as shown in Table 7. For oxygen, phosphate, and nitrate, the bottom boundary conditions are

$$K_H \frac{\partial A_j}{\partial z}\bigg|_{z=z_{\text{end}}} = \lambda_j \left(A_j\big|_{z=z_{\text{end}}} - A_j^*\right), \tag{72}$$

where $\lambda_j$ and $A_j^*$ are the corresponding relaxation velocity and observed at-bottom boundary climatological field data value, respectively, of that species. Base values for the relaxation velocities are included in Table 7. Lastly, the bottom boundary condition for ammonium is dependent on the gradient of particulate organic nitrogen as

$$K_H \frac{\partial N^{(3)}}{\partial z}\bigg|_{z=z_{\text{end}}} = \kappa_{N^{(3)}} \frac{\partial R_N^{(2)}}{\partial z}, \tag{73}$$

where $\kappa_{N^{(3)}}$ is a relaxation diffusivity. In general, ammonium is not often included in observational measurements, so the

gradient in particulate organic nitrogen is used to approximate the bottom boundary condition for ammonium. The relaxation diffusivity for ammonium at the bottom, $\kappa_{N^{(3)}}$, is included in Table 7.

## 4 Field Validation and Calibration Data

### 4.1 Study Site Description

Field data for calibration and validation of BFM17 are taken from the Bermuda Atlantic Time-series Study (BATS) (Steinberg

et al., 2001) and the Bermuda Testbed Mooring (BTM) (Dickey et al., 2001) sites, which are located in the Sargasso Sea





(31°40' N, 64°10' W) in the North Atlantic subtropical gyre. Both sites are a part of the US Joint Global Ocean Flux Study (JGOFS) program. Data has been collected from the BATS site since 1988 and from the BTM site since 1994.

Steinberg et al. (2001) provide an overview of the biogeochemistry in the general BATS and BTM area. Winter mixing allows nutrients to be brought up into the mixed layer, producing a phytoplankton bloom between January and March. As
thermal stratification intensifies over the summer months, this nutrient supply is cut off. At this point, a subsurface chlorophyll maximum is observed near a depth of 100 m. Stoichiometric ratios of carbon, nitrate, and phosphate are often non-Redfield and, in contrast to many oligotrophic regimes, phosphate is the dominant limiting nutrient (Fanning, 1992; Michaels et al., 1993; Cavender-Bares et al., 2001; Steinberg et al., 2001; Ammerman et al., 2003; Martiny et al., 2013; Singh et al., 2015).

### 4.2 Data Processing

The region encompassing the BATS and BTM sites is characterized as an open ocean, oligotrophic region that is phosphate limited. This region has thus been chosen for initial calibration and validation of BFM17 due to the prevalence of oligotrophic regimes in the open ocean and to demonstrate the ability of BFM17 to capture difficult non-Redfield ratio regimes (which occur in phosphate-limited regions). The BATS/BTM data have also been collected over many years, providing long time series for model calibration and validation.

Data from the BATS/BTM area is used in the present study for two purposes: (*i*) as initial, boundary, and forcing conditions for the POM-1D biophysical simulations with BFM17 and BFM56, and (*ii*) as target fields for validation of the simulations. In addition to the subsurface BATS data, we also use BTM surface data, such as the 10 m wind speed and PAR. For each observational quantity, we compute monthly averages over 27 years for the BATS data and 23 years (not continuous) for the BTM data. Additionally, we interpolate the BATS data to a vertical grid with 1m resolution. We subsequently smooth the
interpolated data to maintain a positive buoyancy gradient, thereby eliminating any spurious buoyancy-driven mixing due to interpolation and averaging.

Figure 3 shows the monthly climatological profiles of temperature and salinity from the BATS data, as well as the PAR and 10 m wind speed from the BTM data. Similar processing is also performed on biological variables, which largely serve as target fields for the validation of BFM17. Figure 4 shows monthly-averaged vertical profiles of chlorophyll, oxygen, nitrate,
phosphate, particulate organic nitrogen, and net primary production from the BATS data.

### 4.3 Inputs to the Physical Model

The physical model computes density from the prescribed temperature and salinity, and surface wind stress from the 10 m wind speed; temperature, salinity, and wind speed are all provided by the BATS/BTM data. The model also uses this data in the turbulence closure to compute the turbulent viscosity and diffusivity. This diagnostic approach eliminates any drifts
in temperature and salinity that might occur due to improper parameterizations of lateral mixing in a 1D model, therefore providing greater reliability. In addition to the 10 m wind speed, temperature, and salinity, BFM requires monthly varying PAR at the surface. For all the monthly mean input data sets, a correction (Killworth, 1995) is applied to the monthly averages to account for monthly mean errors due to linear interpolation to the model time step.





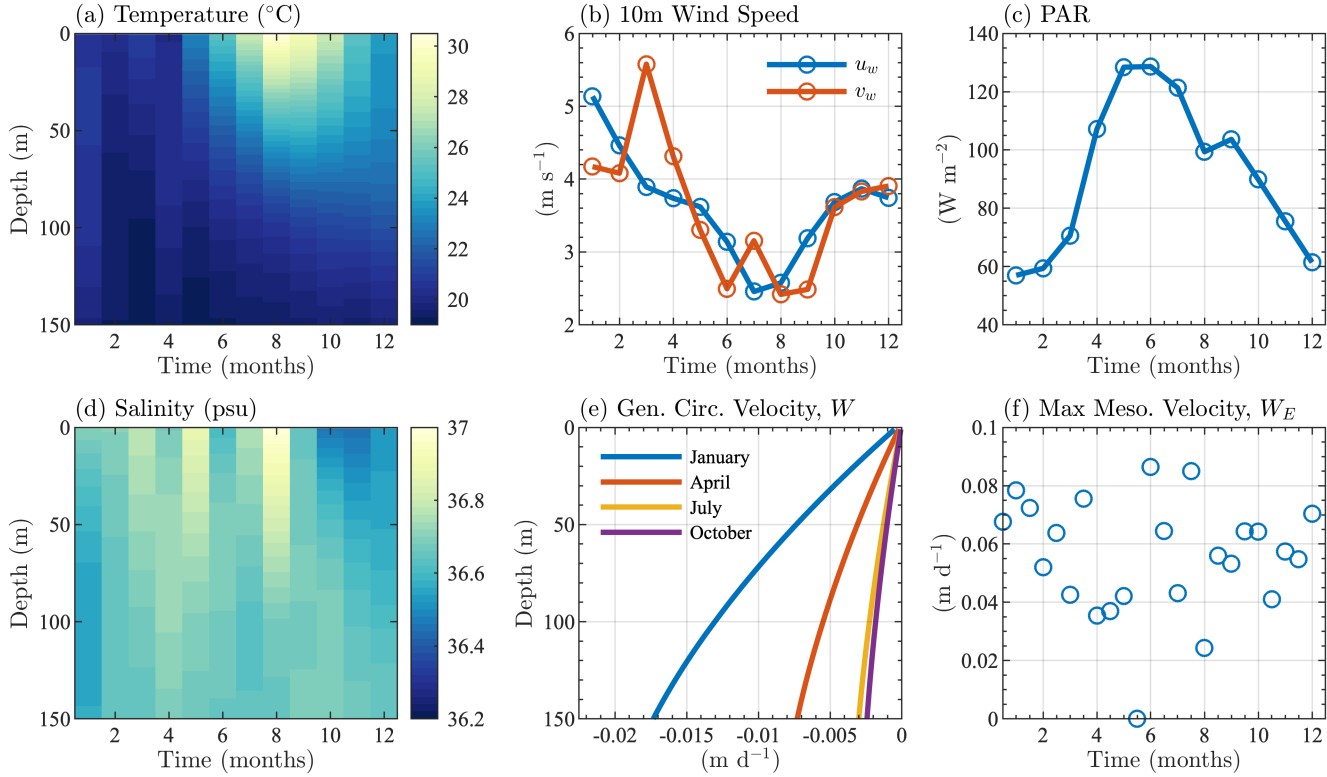

**Figure 3.** Sargasso Sea physical variables, showing climatological monthly averaged (a) temperature, (b) 10 m surface wind speed, (c) surface PAR, and (d) salinity. Panel (e) shows the mean seasonal general circulation velocity, $W$, and panel (f) shows the bimonthly maximum value of the mesoscale eddy velocity $W_E$.

We imposed both general circulation, $W$, and mesoscale eddy, $W_E$, vertical velocities in the simulations. The imposed vertical profiles of these velocities have been adapted from Bianchi et al. (2005), where the the velocities are assumed to be zero at the surface and reach their maxima near the base of the Ekman layer, which is assumed to be at or below the bottom boundary of the simulations. The general large-scale upwelling/downwelling circulation, $W$, is due to Ekman pumping and is correspondingly given as

$$W = \hat{\boldsymbol{k}} \cdot \boldsymbol{\nabla} \times \left( \frac{\boldsymbol{\tau}_w}{\rho f} \right), \tag{74}$$

where $\hat{\boldsymbol{k}}$ denotes the unit vector in the vertical direction. The monthly average value and sign of the wind stress curl, $\boldsymbol{\nabla} \times \boldsymbol{\tau}_w$, for the general BATS/BTM region was taken from the Scatterometer Climatology of Ocean Winds database (Risien and Chelton, 2008, 2011). The monthly value of $W$ from Eq. (74) is then assumed to be the maximum, occurring at the base of the Ekman layer, for that particular month. Given the sign of the wind stress curl for the BATS/BTM region, a negative $W$ was calculated, indicating general downwelling processes in this region. Seasonal profiles of $W$ are shown in Figure 3(e).

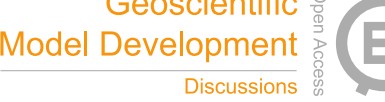

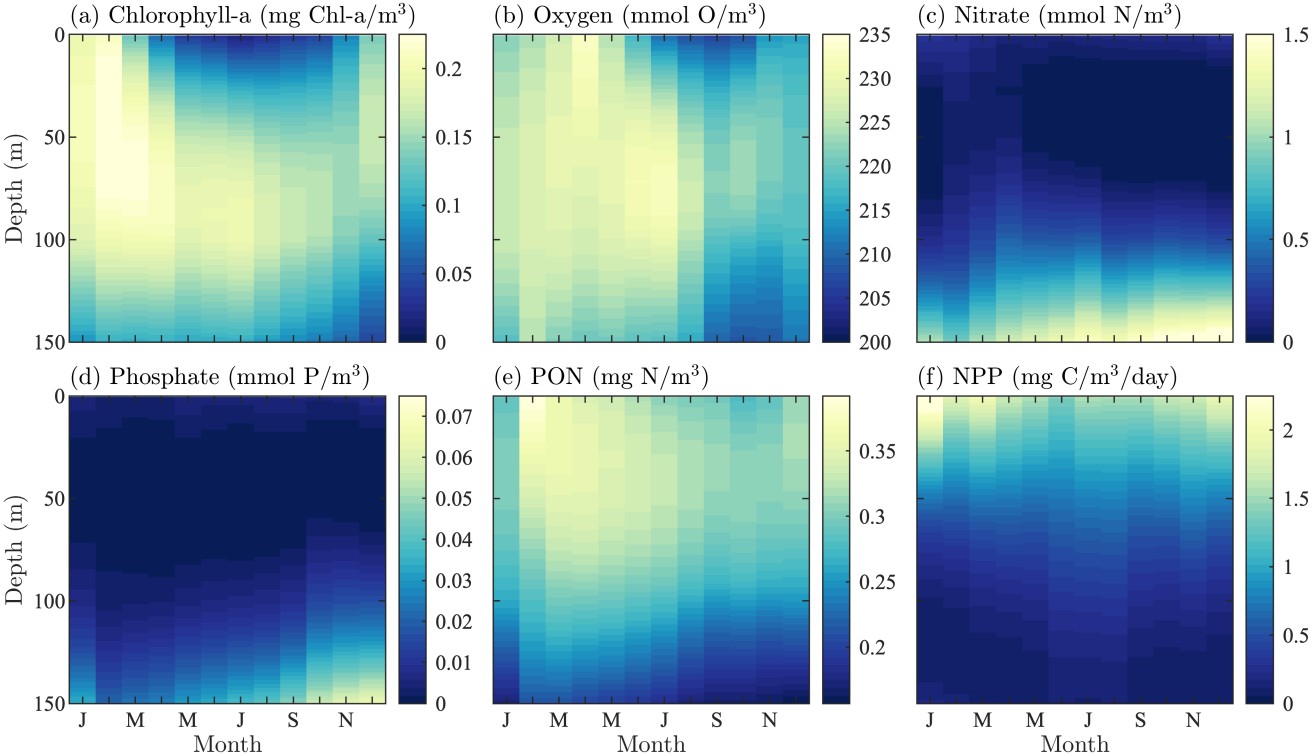

**Figure 4.** Sargasso Sea biological variables, showing climatological monthly averaged (a) chlorophyll-a, (b) oxygen, (c) nitrate, (d) phosphate, (e) particulate organic nitrogen (PON), and (f) net primary production (NPP).

Due to the prevalence of mesoscale eddies within the BATS/BTM region (Hua et al., 1985), which can provide episodic upwelling of nutrients to the upper water column, we also include an additional positive upwelling vertical velocity, $W_E$, which has a timescale of 15 days. The general profile of $W_E$ is assumed to be the same as for $W$, with a value of zero at the surface and a maximum value at depth. However, there is no linear interpolation between each 15-day period and the maximum magnitude of $W_E$ is randomized between 0 and $0.1 \ \mathrm{m \ d^{-1}}$, as shown in Figure 3(f) for each 15-day period.

**4.4    Initial and Boundary Conditions**

Although the BATS/BTM data includes information on many biological variables, initial conditions for only 5 of the 17 species within BFM17 could be extracted from the data. Similarly to the temperature and salinity, the initial chlorophyll, particulate organic nitrogen, oxygen, nitrate, and phosphate were interpolated to a mesh with 1 m vertical grid spacing, averaged over the initial month of January, and smoothed vertically in space to give the initial profiles seen in Figure 5(a). The remaining 12 state
variable initial conditions were determined either through the adoption of the Redfield ratio C:N:P $\equiv$ 106:16:1 (Redfield et al., 2005), or assuming a reasonably low initial value. Since the 1D simulations were run to steady state over 10 years, memory of these initial states was assumed to be lost, with little effect on the results.

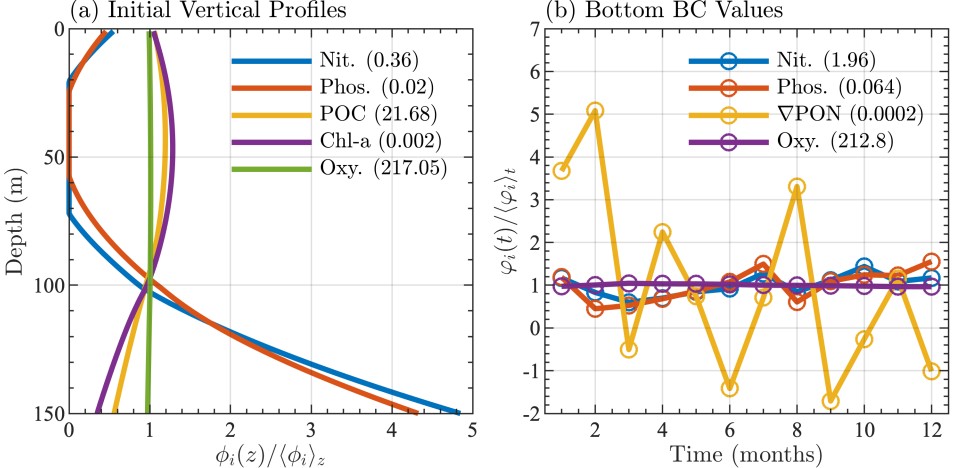

**Figure 5.** Sargasso Sea initial and boundary conditions showing (a) initial profiles of nitrate, phosphate, particulate organic carbon, chlorophyll, and oxygen, where each profile, denoted $\phi_i(z)$, is normalized by its depth averaged value, $\langle\phi_i\rangle_z$, and (b) monthly bottom boundary conditions for nitrate, phosphate, ammonium, and oxygen, where each quantity, $\varphi_i(t)$, is normalized by its annual average value $\langle\varphi_i\rangle_t$. The depth and annual averaged values are shown in parentheses in the legends of each panel. Units are mmol N/m$^3$ for nitrate, mmol P/m$^3$ for phosphatex, mg C/m$^3$ for particulate organic carbon, mg Chl/m$^3$ for chlorophyll, and mmol O/m$^3$ for oxygen.

For the comparison of BFM17 to BFM56, the initial conditions for the additional state variables were calculated by splitting the total phytoplankton and zooplankton carbon values into additional phytoplankton and zooplankton groups. The other state variables for each group were again calculated using the Redfield ratio. The initial bacteria distribution was defined by setting the column equal to a constant value.

In both simulations, the bottom boundary conditions for oxygen, nitrate, phosphate, and ammonium species are based on observed BATS data. For oxygen, nitrate, and phosphate, values are taken at the next closest data point below the bottom boundary (at 150 m) and then averaged over the month. For the ammonium bottom boundary based on Eq. (73), we average over all data points between depths of 125-150 m and 150-175 m across each month for the particulate organic nitrogen, compute the gradient, and then assume the ammonium exhibits a similar gradient. Figure 5(b) shows the monthly average bottom boundary conditions for each of the four species.

## 5 Validation Results

The coupled BFM17-POM1D model was run using the parameter values from Tables 4-7. The empirical values were decided on the basis of standard literature values Vichi et al. (2007, 2003, 2013) with some adjustments to improve agreement with observational data. The simulations were allowed to run out to steady-state and multi-year monthly means were calculated as functions of depth for chlorophyll, oxygen, nitrate, phosphate, particulate organic nitrogen, and net primary production, each of which were measured at the BATS/BTM site.



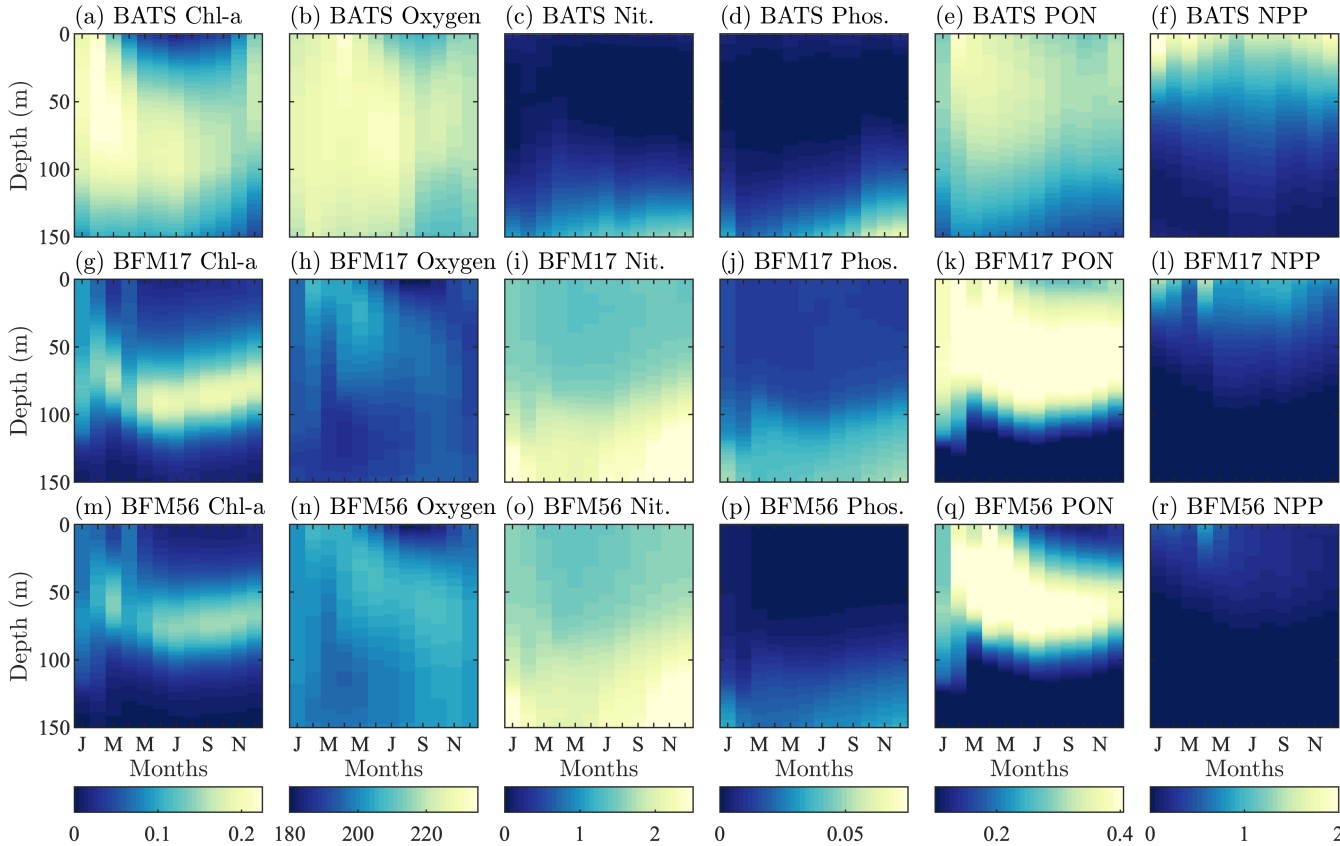

**Figure 6.** Comparison of target BATS fields (top row) to BFM17 simulation results (middle row) and BFM56 simulation results (bottom row) for (a,g,m) chlorophyll (mg Chl-a/m$^3$), (b,h,n) oxygen (mmol O/m$^3$), (c,i,o) nitrate (mmol N/m$^3$), (d,j,p) phosphate (mmol P/m$^3$), (e,k,q) particulate organic nitrogen (PON - mg N/m$^3$), (f,l,r) and net primary production (NPP - mg C/m$^3$/day). Simulation plots are multi-year, monthly averages of the last 3 years of a 10 year integration.

Figure 6 qualitatively compares the BATS data (top row) with the results of from BFM17 (middle row). The model is able to
capture the initial spring bloom between January and March brought on by physical entrainment of nutrients, the corresponding peak in net primary production and PON around the same time, and the subsequent subsurface chlorophyll maxima during the summer. The predicted oxygen levels are lower than observed values, while nitrate and phosphate levels are generally higher than observed values. The overall structures of oxygen, nitrate, and phosphate predicted by BFM17 are, however, quite similar to that of the BATS target fields. These trends are consistent with the trends seen in the results from BFM56 (bottom row of
Figure 6), suggesting that the two models are in generally close agreement.

To quantitatively evaluate BFM17, a model skill assessment was performed for each target field. The same skill assessment was performed for BFM56 to compare the two models. The results are summarized by the Taylor diagram in Figure 7. This diagram can be used to assess the extent of misfit between the models and observations by showing the normalized root mean



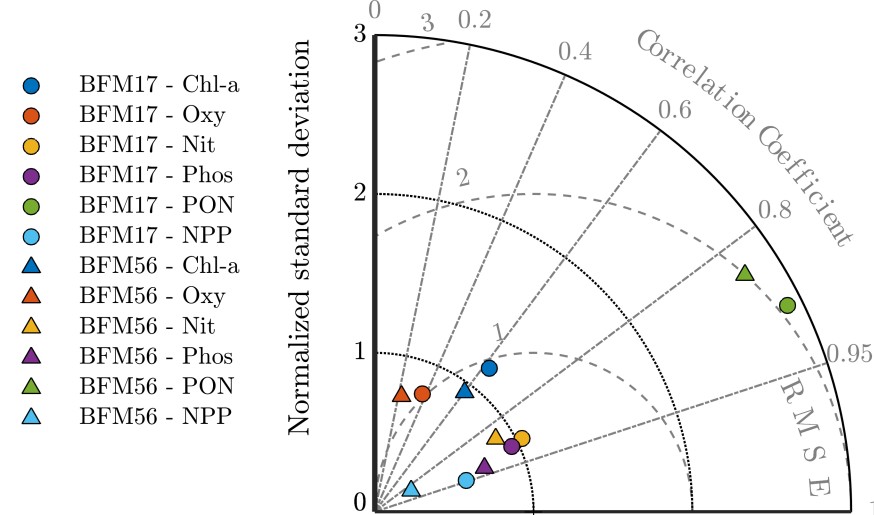

**Figure 7.** Taylor diagram showing the normalized standard deviation, correlation coefficient, and normalized root mean squared differences between the BFM17 output and the BATS target fields. Observations lie at (1,0). Radial deviations from observations corresponds to the normalized root mean square error (RMSE), radial deviation from the origin correspond to the normalized standard deviation, and angular deviations from the vertical axis correspond to the correlation coefficient. BFM17 and BFM56 results are shown as colored circles and triangles, respectively (chlorophyll = blue, oxygen = orange, nitrate = yellow, phosphate = purple, PON = green, NPP = cyan).

square (RMS) errors, normalized standard deviation, and the correlation coefficient between each of the model outputs and the

BATS target fields.

The normalized RMS errors were calculated as $\varepsilon_{\mathrm{rms}}/\sigma_{\mathrm{obs}}$, where $\varepsilon_{\mathrm{rms}}$ is the RMS error between the model and the observation fields and $\sigma_{\mathrm{obs}}$ is the standard deviation of the observation field. The normalized standard deviation was calculated as $\sigma_{\mathrm{mod}}/\sigma_{\mathrm{obs}}$ where $\sigma_{\mathrm{mod}}$ is the standard deviation of the model fields. The normalized RMS errors, normalized standard deviation, and the correlation coefficients each give an indication of the relative similarities in amplitude, variations in amplitude,

and structure of each modeled field compared to the BATS target fields, respectively. For each variable, these statistics were calculated over all months and all depths shown in Figure 6.

The Taylor diagram in Figure 7 shows that BFM17 and BFM56 produce similar results. For all variables except oxygen, errors in the amplitudes are within roughly one standard deviation of the observations. Additionally, the structure of the model fields for chlorophyll, nitrate, phosphate, PON, and NPP have high correlations with that of the BATS target fields. The corre-

lation values range from 0.62 for chlorophyll to 0.95 for NPP in BFM17 and from 0.60 for chlorophyll to 0.93 for phosphate in BFM56. The variability in amplitude for chlorophyll, nitrate, and phosphate are closest to that of the corresponding BATS target fields, while the oxygen and NPP have a relative lack of variability, and PON has too much variability.





**Table 8.** Correlation coefficients (and RMS error in parenthesis) between BATS target fields and model data for BFM17, BFM56, and several example reduced-order models.

| Variable | BFM17 | BFM56 | Ayata *et al.* (2013) | Fasham *et al.* (1990) | Spitz *et al.* (2001) |
|---|---|---|---|---|---|
| Chlorophyll | 0.62 (0.08) | 0.60 (0.10) | 0.60 (0.06) | -0.33 (0.34) | 0.86 (0.04) |
| Oxygen | 0.37 (29.59) | 0.23 (22.43) | - | - | - |
| Nitrate | 0.90 (1.47) | 0.86 (1.51) | 0.80 (0.33) | 0.87 (0.28) | 0.98 (0.05) |
| Phosphate | 0.90 (0.01) | 0.93 (0.01) | - | - | - |
| PON | 0.89 (0.10) | 0.84 (0.11) | 0.45 (0.08) | 0.48 (0.6) | 0.76 (0.12) |
| NPP | 0.95 (0.43) | 0.85 (0.63) | 0.50 (0.14) | -0.47 (0.021) | 0.69 (0.016) |

Table 8 provides a comparison of correlation coefficients and un-normalized RMS errors from BFM17 and BFM56, as well as from other models. Comparisons were only made to models that were calibrated using the same BATS/BTM data, employed some kind of parameter estimation technique, were forced with a similar one-dimensional physical model, and reported correlation and RMS errors. Ayata et al. (2013) contained six biological tracers, while both Fasham et al. (1990) and Spitz et al. (2001) contained seven. The Spitz et al. (2001) study used data assimilation, while the Ayata et al. (2013) and Fasham et al. (1990) studies used only optimization to determine a select set of parameters.

Table 8 shows that BFM17 and BFM56 give comparable results, with BFM17 producing similar correlation coefficients and RMS error values to those from BFM56. The largest differences compared to the BATS data for both BFM17 and BFM56 are in the oxygen values. The overall slightly better agreement of BFM17 with the BATS data results from the adjustment of some model parameters in BFM17 due to the removal of specific phytoplankton and zooplankton species in favor of general LFGs, and to the parameterization of remineralization using new closure terms that were calibrated to give reasonable agreement with the observational data. By contrast, BFM56 was run using the baseline parameter values that were not adjusted to improve agreement with the observations.

The correlation coefficients and RMS errors for both BFM17 and BFM56 are also comparable with the Ayata et al. (2013) and Fasham et al. (1990) studies for chlorophyll and nitrate, while out-performing these studies for PON and NPP. The Spitz et al. (2001) study, which used data assimilation and is therefore naturally more likely to perform better, does in fact do so for predictions of chlorophyll and nitrate. However, the nitrate correlation values for BFM17 and the Spitz et al. (2001) model are both high, although the latter model does have a lower RMS error value. As compared to the Spitz et al. (2001) model, BFM17 has higher correlation values for both PON and NPP, but a larger RMS error for NPP.

These results show that, with a relatively small increase in the number of biological tracers as compared to similar models, BFM17 is generally able to increase correlation coefficient values and decrease RMS error values for each target field in comparison to similar models. Moreover, BFM17 approaches the accuracy of models that use data assimilation to improve agreement with the observations, such as the Spitz et al. (2001) model. The extra biological tracers in BFM17, as compared to the Ayata et al. (2013) and Fasham et al. (1990) models, account for variable intra- and extra-cellular nutrient ratios with the addition of phosphorus



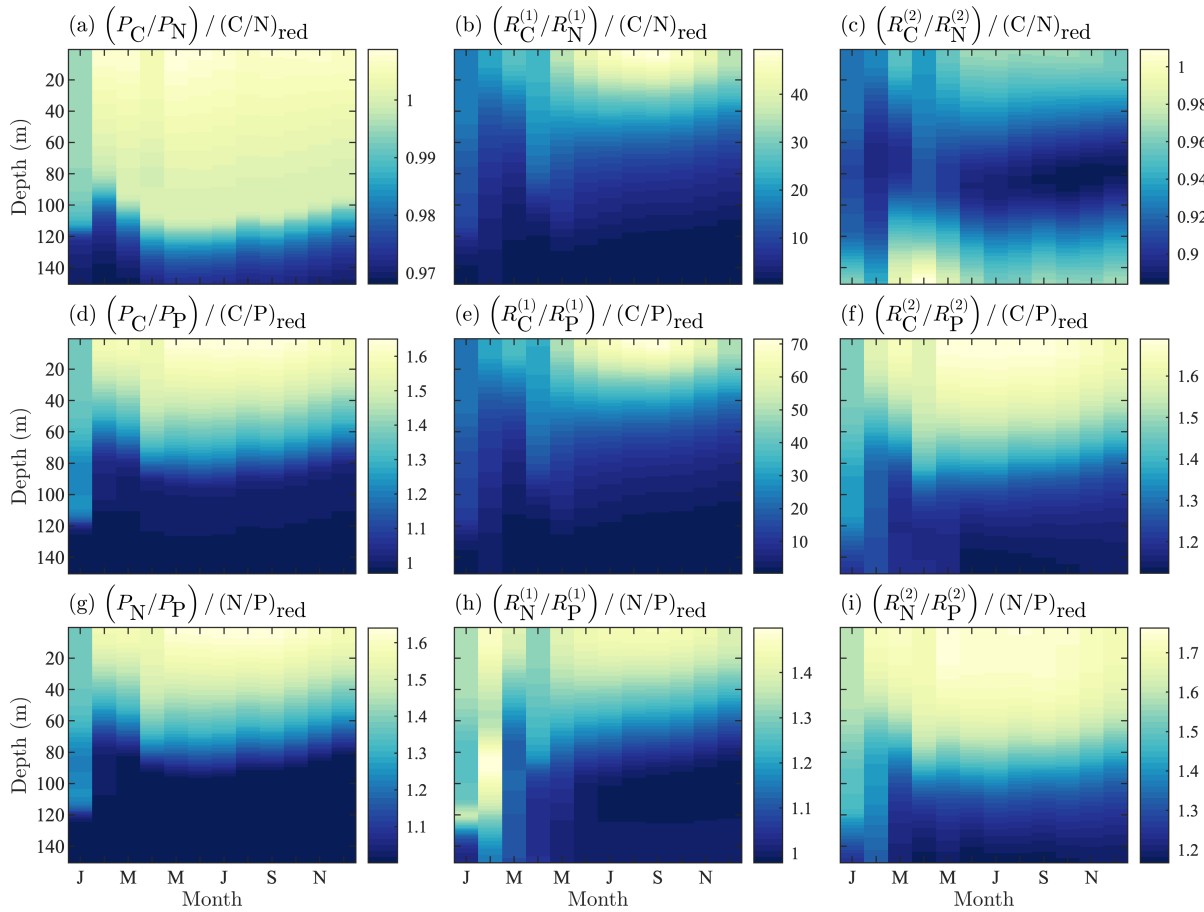

**Figure 8.** Fields of BFM17 constituent component ratios of carbon to nitrogen (top row), carbon to phosphorous (middle row), and nitrogen to phosphorous (bottom row) for phytoplankton (first column), dissolved organic detritus (second column), and particulate organic detritus (third column). Each field is normalized by the respective Redfield ratio.

Finally, a key benefit of the chemical functional family approach used by BFM17 is the ability of the model to predict non-Redfield nutrient ratios. Figure 8 shows the constituent component ratios normalized by the respective Redfield ratios for BFM17. The figure includes the component ratios of carbon to nitrogen, carbon to phosphorous, and nitrogen to phosphorous for phytoplankton, DOM, and POM. Zooplankton nutrient ratios were not included because the parameterization of the zooplankton relaxes the nutrient ratio back to a constant value. The normalized ratio values are uniform non-unity valued fields.

Ultimately, Figure 8 shows that BFM17 is able to capture the phosphate-limited dynamics that characterize the BATS/BTM region (Fanning, 1992; Michaels et al., 1993; Cavender-Bares et al., 2001; Steinberg et al., 2001; Ammerman et al., 2003; Martiny et al., 2013; Singh et al., 2015). In particular, Figure 8 shows that all results comparing carbon or nitrogen to phosphorous for BFM17 produce normalized vales greater than 1, where the normalization is carried out using the Redfield ratio (i.e., a normalized value greater than 1 indicates that the field is denominator limited). Figure 8 also shows that the ratios are



not uniform for phytoplankton, DOM, and POM, with the ratios decreasing with depth as a result of the increased availability of nitrogen and phosphate.

## 6 Conclusions

In this study, we have presented a new reduced-order BGC model that is complex enough to capture open-ocean ecosystem dynamics within the Sargasso Sea region, yet reduced enough to integrate with a physical model with limited additional computational cost. The new model, named the Biogeochemical Flux Model 17 (BFM17) includes 17 state variables and expands upon more reduced BGC models by incorporating a phosphate equation, as well as the ability to track variable intra- and extra-cellular nutrient ratios.

To calibrate and test the model, it was coupled to the one-dimensional Princeton Ocean Model (POM-1D) and forced using field data from the Bermuda Atlantic test site area. The full 56 state variable Biogeochemical Flux Model (BFM56) was also run using the same forcing. Results were compared between the two models and all six of the BATS target fields—chlorophyll, oxygen, nitrate, phosphate, PON, and NPP—and a model skill assessment was performed, concluding that the BFM17 does well at reproducing observations and produces comparable results to BFM56. In comparison with similar studies using slightly less complex models, BFM17-POM1D performs on par with, or better than, those studies.

In the future, a sensitivity study is necessary to assess the most sensitive model parameters, both in BFM17 as well as in the 1D physical model. After identification of these most sensitive model parameters, an optimization can be performed to reduce discrepancies between the BATS observation biology fields and the corresponding model output fields. Finally, BFM17 is now of a size that it can be efficiently integrated in a large-scale LES, and future work will examine model results in 3D simulations of the upper ocean.

*Code and data availability.*  Current versions of BFM17 and BFM56 are at https://github.com/marco-zavatarelli/BFM17-56/tree/BFM17-56 under the GNU General Public License version 3. The exact versions of the models used to produce the results used in this paper are archived on Zenodo at http://doi.org/10.5281/zenodo.3839984. Data and scripts used to produce all figures in this paper are archived on Zenodo at http://doi.org/10.5281/zenodo.3840562.

*Author contributions.*  KMS, SK, PEH, MZ, and NP formulated the reduced order model, EFK and KEN verified and debugged the model, KMS and SK ran the model simulations and produced the results presented in the paper, KMS and PEH prepared the initial draft of the paper, and all authors edited the paper to produce the final version.

*Competing interests.*  The authors declare that they have no competing interests or other conflicts of interest.



*Acknowledgements.* KMS and PEH were supported by NSF OCE-1258995 and PEH was supported, in part, by NSF OCE-1924636. SK was supported by an ANSEP Alaska Grown Fellowship and by an NSF Graduate Research Fellowship. EFK and KEN were supported by NSF OAC-1535065. The data analyzed in this paper are available from Mendeley Data (https://data.mendeley.com). This work utilized the RMACC Summit supercomputer supported by NSF (ACI-1532235, ACI-1532236), CU-Boulder, and CSU, as well as the Yellowstone (ark:/85065/d7wd3xhc) and Cheyenne (doi:10.5065/D6RX99HX) supercomputers provided by NCAR CISL, sponsored by NSF.





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
