# Peer review of "BFM17 v1.0: A Reduced Biogeochemical Flux Model for Upper Ocean Biophysical Simulations"

_Geoscientific Model Development, 2020_

## Referee Comment (RC1) · Anonymous Referee #1 · 2 Sep 2020

Referee Report on Manuscript

GMD-2020-134

"BFM17 v1.0: Reduced-Order Biogeochemical Flux Model for Upper Ocean Biophysical Simulations"

By

Katherine M. Smith et al.

This manuscript concerns the development of a biogeochemical model of open-ocean ecosystem dynamics called BFM17. The authors make the case that the model is

"complex and flexible enough to capture open-ocean ecosystem dynamics, but reduced enough to incorporate into highly resolved numerical simulations with limited additional computational cost.". Furthermore, they provide validation of the model in 0D and 1D mode with measured data from an oligotrophic open ocean station.

In general, I agree with the claims of the authors. My view is that reduced complexity models such as this one are important for process studies, so I welcome the initiative of the authors to develop one more option for biogeochemical modelers. It would be preferable to have a validation also in a 3d setting, which is the one that the model was developed for but, since the model equations use standard formulations for the source and sink terms, I believe that the validation presented in the manuscript is sufficient.

Overall, I think the model has the potential to become a valuable addition to the tools available to study ocean biogeochemistry processes. With models such as these, only time can tell if it will fulfill this promise. My only remark is that, like similar papers, little time is dedicated to describe the way the parameter values are found. I find it peculiar that since, as everybody in this business knows, model parameters are paramount to getting decent model results, hardly no one takes the time to explain how they got to the values they are using. Thus, I urge the authors to add a little more detail on this matter, because it will certainly help other that may want to use the BFM17 model in their work.

Please check the reference style in line 347.

---

## Referee Comment (RC2) · Anonymous Referee #2 · 21 Oct 2020

Review on "BFM17 v1.0: Reduced-Order Biogeochemical Flux Model for Upper Ocean Biophysical Simulations" by Katherine M. Smith, Skyler Kern, Peter E. Hamlington, Marco Zavatarelli, Nadia Pinardi, Emily F. Klee, and Kyle E. Niemeyer

Summary: The presented manuscript compares two model versions of a pelagic biogeochemical model, embedded in a 1-dimensional ocean model. The first model version refers to the BFM-model with 56 state variables (BFM56), while a second, reduced version (BFM17) includes only 17 state variables for the biogeochemistry. The authors compare the results of both model versions to climatological observations at the sites Bermuda Atlantic Timeseries (BATS) and Bermuda Testbed Mooring (BTM).

[Figure]

Major comments: The manuscript addresses the important, so far unresolved question on the necessary complexity of pelagic biogeochemical models. The topic is per se interesting. Still I have some major concerns:

(1) BFM17 excludes some processes relative to BFM56. This reduces by construction the applicability of the newly developed model (e.g., coastal shallow regions might not be represented so well anymore, due to the lack of the sediment processes, while simulating global phosphate distributions might still work very well). I would thus appreciate a clear statement for which purposes the new model was designed. Also, the overall aim of the presented results did not get entirely clear to me: is the idea of BMF17 to mimic/resemble certain processes of BFM56 or is BFM17 a model on its own, which might perform even better than BFM56 relative to the observations?

(2) In my eyes the study would need a better illustration that the presented model is up to the tasks it is designed for – or at least state-of-the-art. Currently, both model versions are compared to climatological, station based in-situ observations close to Bermuda. I am surprised that both model versions don't do better (e.g., for oxygen the explained variances are in the order of 5-15%, cf. Fig.7). The study might get more convincing if the authors could at least outline what goes wrong.

(3) When reading the model description, I found it rather difficult to keep overview what exactly was done. I would suggest to summarize the main model assumptions somewhere in the beginning of the model description (the equations might as well be collectively listed in an appendix). E.g. it did not become clear to me how the reduction of state variables was motivated and how the new state variables and parameter settings were chosen.

(4) I agree in general with the approach to compare both model versions initially in a low-dimensional environment, which is less costly to run than a full-fledged ocean model. Unfortunately, the authors do not take advantage of such a computationally cheap setup, e.g., by testing the sensitivities of both models towards environmental

changes in idealized experiments. Such an approach would be feasible without too much effort and could explore how much BFM17 resembles BFM56 under various conditions (preferably also in well fertilized regions).

I am, however, sceptic about comparing the 1-dimensional model results to the observations. First, the comparison captures only one very special region – while I assume that the model is rather designed to answer questions on a global scale and I regard a 3-dimensional ocean component as more appropriate. Secondly, I fear that ocean mixing might not be realistically represented and that the restoring below 150m might induce spurious effects in the 1-dimensional model. In the real ocean, regions with intense sinking of organic matter and upwelling of nutrients are often spatially decoupled. How I understood the presented setup, the 1-dimensional model needs to balance both processes in a single box and thus induces some eddy driven transports (line 504ff + line 515ff), to reach some kind of steady-state (line 526). The latter seems rather arbitrary to me (please correct me in case I got things wrong). My scepticism is supported by the rather poor representation of oxygen in both model versions (because oceanic oxygen concentrations close to the surface are strongly impacted by water temperature and mixing). Also, I am puzzled that both models deviate strongly from the observations even at 150m – i.e. close to the prescribed boundary conditions at depth (cf., Fig. 6).

Minor comments:

Line 1: I find the expression "reduced-order model" in the presented context a bit confusing. The authors seem to use this term whenever a certain (non-defined) number of state variables is undercut. It might make sense to define the expression somewhere.

Line 7: I expect large differences in the tropical Pacific when disregarding iron in BFM17 (unless the top-down control is too strong). Depending on the region of interest the authors might consider to use at least a fixed, prescribed iron concentrations.

Line 79: The "accuracy" certainly depends on the underlying measure. What is regarded as important will certainly depend on the selected region and the purpose of the model.

Line 83ff: So which parameters were calibrated and how was this done?

Line 135ff: I agree that presenting all equations is important. Still I find it impossible to keep overview in the presented form. The rationale behind the construction of the new BFM17-model does not at all become clear to me.

Line 380ff: I find it very hard to understand the physical setup and would appreciate if the main assumptions were summarized in the beginning of this Subsection. Currently, I get lost in too many details. Also, it should be clearly stated that only the upper 150m are simulated. What is the maximum mixed layer depth?

Line 520: I would suggest to describe initial values and boundary conditions in relation to the description of the physical setup.

Figure 2 What is the purpose of this figure? Why are not both model versions depicted and what do these experiments add to the conclusions?

Figure 3+4: Again – the purpose of showing these figures does not become clear to me (e.g., Fig.4 vs. Fig.6?)

Figure 6: I see substantial differences between both model versions – even in the base currency "phosphate". How do the authors rate which differences matter? Why are the models so different from the observations at 150m (close to the bottom boundary conditions)?

Figure 7: Compared to the huge differences between the reference model and the observations, both model versions seem indeed rather similar (while in fact I regard these differences as substantial). I would suggest to add a more intuitive reprsentation of the differences between both model versions in the corresponding text, e.g., % relative to the reference model.

Table 8 + line 568ff: I like the idea behind this comparison to illustrate that the presented model is state-of-the-art. Still comparability needs to be ensured. Do all cited models refer to the same sampling in time (please specify) and state variables? Otherwise these numbers are not comparable. Also, the underlying physical model setups should be listed as well. It is much harder to simulate a specific timeseries in a 3D context than to be able to tune the model at a single location (looking at Figure 1 and the tables, I would expect at least 40 poorly known parameters to match 12 moments in time – this should work perfectly at least for one state variable at one depth level).

Line 605ff (Conclusion): I would suggest to add a clear statement for which purpose the model BFM17 was designed and what it could be used for. Subsequently, the study would in my eyes benefit from an outline why the authors think that the model is up-to-the-task, or at least state-of-the-art.

Line 615: As outlined before, I do not agree on the generalized, conclusive statement "the model does well".

$\sim$

---

## Author Comment (AC1) · 17 Dec 2020

**Response to Reviewer 1**

We appreciate the constructive and insightful comments from the reviewer. A number of changes have been made to the text at the reviewer's request and we feel that the revised paper is now substantially clearer and stronger as a result. Detailed replies to each of the reviewer's points (in italics) are provided below.

*This manuscript concerns the development of a biogeochemical model of open-ocean ecosystem dynamics called BFM17. The authors make the case that the model is*

[Figure]

*"complex and flexible enough to capture open-ocean ecosystem dynamics, but reduced enough to incorporate into highly resolved numerical simulations with limited additional computational cost." Furthermore, they provide validation of the model in 0D and 1D mode with measured data from an oligotrophic open ocean station. In general, I agree with the claims of the authors. My view is that reduced complexity models such as this one are important for process studies, so I welcome the initiative of the authors to develop one more option for biogeochemical modelers. It would be preferable to have a validation also in a 3D setting, which is the one that the model was developed for but, since the model equations use standard formulations for the source and sink terms, I believe that the validation presented in the manuscript is sufficient. Overall, I think the model has the potential to become a valuable addition to the tools available to study ocean biogeochemistry processes. With models such as these, only time can tell if it will fulfill this promise.*

**Comments:**

*My only remark is that, like similar papers, little time is dedicated to describe the way the parameter values are found. I find it peculiar that since, as everybody in this business knows, model parameters are paramount to getting decent model results, hardly no one takes the time to explain how they got to the values they are using. Thus, I urge the authors to add a little more detail on this matter, because it will certainly help other that may want to use the BFM17 model in their work.*

We agree with the reviewer and we will provide an explanation of how the parameter values were determined in the revised manuscript. Briefly, the values used in this paper were based largely on the values established by Vichi *et al.* (2003, 2007, 2013) for BFM56. We are also currently pursuing computational optimization of these parameters to minimize model error when compared to the BATS observational data; this future work will be mentioned in the Conclusions section

of the manuscript.

*Please check the reference style in line 347.*

Thank you for catching this mistake. The correct reference style will be used in the revised manuscript.

---

## Author Comment (AC2) · 17 Dec 2020

**Response to Reviewer 2**

We appreciate the constructive and insightful comments from the reviewer. A number of changes have been made to the text at the reviewer's request and we feel that the revised paper is now substantially clearer and stronger as a result. Detailed replies to each of the reviewer's points (in blue italics) are provided below.

*The presented manuscript compares two model versions of a pelagic biogeochemical model, embedded in a 1-dimensional ocean model. The first model version refers to the*

*BFM-model with 56 state variables (BFM56), while a second, reduced version (BFM17) includes only 17 state variables for the biogeochemistry. The authors compare the results of both model versions to climatological observations at the sites Bermuda Atlantic Timeseries (BATS) and Bermuda Testbed Mooring (BTM).*

*The manuscript addresses the important, so far unresolved question on the necessary complexity of pelagic biogeochemical models. The topic is per se interesting. Still I have some major concerns.*

*Major Comments:*

*BFM17 excludes some processes relative to BFM56. This reduces by construction the applicability of the newly developed model (e.g., coastal shallow regions might not be represented so well anymore, due to the lack of the sediment processes, while simulating global phosphate distributions might still work very well). I would thus appreciate a clear statement for which purposes the new model was designed.*

We agree with the reviewer that the applicability of BFM17 is reduced in comparison to BFM56 and, in order to more clearly state the range of applicability of the model, we will state in the revised manuscript that "BFM17 is a pelagic model intended for upper-thermocline, open-ocean, non-iron or silicate limited, oligotrophic regions". With respect to the purpose of the model, we will clarify that "BFM17 has been developed for the primary intended use within high-resolution, high-fidelity idealized models, such as a Large Eddy Simulation (LES), for process, parameterization, and optimal parameter estimation studies (BFM56 is far too costly for this kind of application). Thus we do not validate its efficacy as a global BGC model (though its relative agreement with BFM56 suggests its potential to do quite well as such) and provide the caveat that it is missing potentially important processes for such an application."

*Also, the overall aim of the presented results did not get entirely clear to me: is the idea of BMF17 to mimic/resemble certain processes of BFM56 or is BFM17 a model on its own, which might perform even better than BFM56 relative to the observations?*

We agree with the reviewer that this was not entirely clear and we will add the following text to the revised manuscript: "Additionally, we later compare to the original BFM56, not in an effort to demonstrate that BFM17 is superior and thus should be used in place of BFM56 in global applications, but rather to demonstrate that, although it is reduced in complexity, BFM17 is equally appropriate for use in seasonal process, parameterization, and optimal parameter estimation studies for which a more complex model such as BFM56 may be too computationally expensive."

*In my eyes the study would need a better illustration that the presented model is up to the tasks it is designed for—-or at least state-of-the-art. Currently, both model versions are compared to climatological, station based in-situ observations close to Bermuda. I am surprised that both model versions don't do better (e.g., for oxygen the explained variances are in the order of 5-15%, cf. Fig.7). The study might get more convincing if the authors could at least outline what goes wrong.*

We agree with the reviewer that not all presented fields are reproduced by BFM17 and BFM56 with great accuracy (e.g., oxygen), and we will clarify that the exact reproduction of all fields was not our goal with BFM17. In the revised manuscript, we will state that "...oxygen is historically a difficult field to reproduce in BGC models of any complexity. It is hypothesized that this is due in part to a lack of physical processes being represented in mixing parameterizations. For example, our model struggles with oxygen, in part, because the second-order mixing scheme of Mellor and Yamada (1982) lacks sufficient resolution of the winter mixing using just the monthly mean temperature and salinity. However, since it is often not included or presented at all in models of similar complexity to BFM17 (i.e.

models reduced enough to reasonably couple to a high-fidelity, high-resolution physical model), studies that explore this hypothesis have been historically difficult to undertake. Thus, we include oxygen in BFM17 and present our results here to illustrate this exact point and lend motivation to developing and using a model such as BFM17 to study what physical processes are missing from mixing parameterizations and how they can be better represented."

*When reading the model description, I found it rather difficult to keep overview what exactly was done. I would suggest to summarize the main model assumptions somewhere in the beginning of the model description (the equations might as well be collectively listed in an appendix). E.g. it did not become clear to me how the reduction of state variables was motivated and how the new state variables and parameter settings were chosen.*

We agree that our previous model description was potentially difficult to follow. We will thus move the detailed equations and 0D test to appendices in the revised manuscript. This will help to focus the reader in the main model description section on the model assumptions and choice of state variables.

*I agree in general with the approach to compare both model versions initially in a low-dimensional environment, which is less costly to run than a full-fledged ocean model. Unfortunately, the authors do not take advantage of such a computationally cheap setup, e.g., by testing the sensitivities of both models towards environmental changes in idealized experiments. Such an approach would be feasible without too much effort and could explore how much BFM17 resembles BFM56 under various conditions (preferably also in well fertilized regions).*

We agree with the reviewer that this kind of comparison would be computationally cheap and easy to perform. However, since BFM17 was primarily developed for use in high-resolution, high-fidelity physical models (something BFM56 cannot be

used for because of its cost) and not for use as a global or column model, this comparison is beyond the scope of the model description and baseline demonstration that are the primary objectives of the present paper. Nevertheless, we do agree with the reviewer that this is an important direction for future research, and we will clarify this point in the revised manuscript with the following text: "It should be noted that the primary focus in the present study is to demonstrate the efficacy of BFM17 as an accurate BGC model in high-resolution, high-fidelity upper-ocean process, parameterization, or optimal parameter estimation studies, by comparing to results from observations and BFM56; as such, here we only consider one open-ocean location (i.e., the Sargasso Sea) and make the assumption that its ability to reproduce important and difficult key behaviors at this location, such as the initial spring bloom and subsequent subsurface chlorophyll maxima, are indications of its efficacy for use as a process study model. However, the correspondence between BFM17 and the more general BFM56 provides confidence that the reduced model will also prove effective at modeling other ocean locations and conditions, and exploring the range of applicability of BFM17 remains a subject of future research."

*I am, however, sceptic about comparing the 1-dimensional model results to the observations. First, the comparison captures only one very special region – while I assume that the model is rather designed to answer questions on a global scale and I regard a 3-dimensional ocean component as more appropriate.*

We apologise for any confusion we might have caused by not clearly stating that BFM17 was not developed for use as a global model. We will clearly state this in the revised manuscript: "BFM17 has been developed for the primary intended use within high-resolution, high-fidelity for process, parameterization, and optimal parameter estimation studies (BFM56 is far too costly for this kind of application). Thus we do not validate its efficacy as a global BGC model (although its relative

agreement with BFM56 suggests its potential to do quite well as such) and note that it is missing potentially important processes for such an application, which we elaborate on shortly."

*Secondly, I fear that ocean mixing might not be realistically represented and that the restoring below 150m might induce spurious effects in the 1-dimensional model. In the real ocean, regions with intense sinking of organic matter and upwelling of nutrients are often spatially decoupled. How I understood the presented setup, the 1-dimensional model needs to balance both processes in a single box and thus induces some eddy driven transports (line 504 + line 515), to reach some kind of steady-state (line 526). The latter seems rather arbitrary to me (please correct me in case I got things wrong). My scepticism is supported by the rather poor representation of oxygen in both model versions (because oceanic oxygen concentrations close to the surface are strongly impacted by water temperature and mixing).*

We agree with the reviewer that any 1D column model is unlikely to realistically represent all processes, particularly those associated with mixing and spatially decoupled processes. However, the 1D physical model is only used here for validation purposes, and a primary reason for developing BFM17 is to explore issues such as the effects of mixing and temperature in the surface ocean on oxygen or how "patchy" (or spatially decoupled) nutrient upwelling effects the distribution and fate of a phytoplankton bloom using more physically realistic fully non-hydrostatic 3D simulations. We will elaborate on this in the text of the revised manuscript through the following additions: "Dissolved oxygen is of particular interest as it is historically difficult to get correct using BGC models of any complexity. It is hypothesized that this is in part due to missing physical processes in the mixing parameterizations used in global and column models. A primary goal of developing BFM17 is to use it within high-resolution, high-fidelity physical models to understand what these physical processes are and how to properly account for

them in mixing parameterizations. Dissolved and particulate organic matter, each with their own partitions of carbon, nitrogen, and phosphate, are also included to account for nutrient recycling and carbon export due to particle sinking. Another primary goal of developing BFM17 is to explore how spatially decoupled (or "patchy") processes, such as the sinking of organic matter and the subsequent upwelling of multiple recycled nutrients, not just nitrate, effect the fate and distribution of a phytoplankton bloom."

*Also, I am puzzled that both models deviate strongly from the observations even at 150m – i.e. close to the prescribed boundary conditions at depth (cf., Fig. 6).*

We agree with the reviewer that the nutrient profiles (particularly nitrate) deviate strongly from the observations at depth. Initially, in order to represent the recycling of PON and surface layer nitrogen replenishment from depth, we included a bottom boundary condition for ammonium based on the gradient in PON observations (direct observations of ammonium at the BATS location were lacking). However, as the reviewer points out, this process can be spatially decoupled and thus the assumptions we made associated with this boundary condition are potentially unsupported. In light of the fact that we do not know the exact relationship between the sinking PON and up-welled ammonium, we have now chosen to set this boundary condition to zero (consistent with the idea that most of the nitrogen flux from depth is occurring as nitrate flux). The nutrient results for both models now more closely match observations, both overall and at depth. The new correlation coefficients and *rms* errors in parentheses are now as follows for BFM17: Chlorophyll = 0.63 (0.08), Oxygen = 0.37 (31.18), Nitrate = 0.94 (0.22), Phosphate = 0.91 (0.01), PON = 0.85 (0.15), NPP = 0.93 (0.26). For BFM56, they are: Chlorophyll = 0.60 (0.10), Oxygen = 0.18 (21.84), Nitrate = 0.93 (0.16), Phosphate = 0.93 (0.005), PON = 0.85 (0.11), NPP = 0.87 (0.63).

***Minor Comments:***

*Line 1: I find the expression "reduced-order model" in the presented context a bit confusing. The authors seem to use this term whenever a certain (non-defined) number of state variables is undercut. It might make sense to define the expression somewhere.*

We agree that this expression is confusing. We will remove this description and replace it with "upper-thermocline, open-ocean" throughout the revised manuscript in order to be clear as to how exactly the model is "reduced".

*Line 7: I expect large differences in the tropical Pacific when disregarding iron in BFM17 (unless the top-down control is too strong). Depending on the region of interest the authors might consider to use at least a fixed, prescribed iron concentrations.*

We agree with the reviewer that BFM17 may not perform well in regions where iron-limitation is important, such as the tropical Pacific and the Southern Ocean. We believe that the following altered statements in the revised manuscript convey that BFM17 was not developed, nor has it been qualified, for used in such regions: "To reduce the overall computational cost and to focus on upper-thermocline, open-ocean, and non-iron or silicate limited conditions..." and "BFM17 is a pelagic model intended for upper-thermocline, open-ocean, non-iron or silicate limited, oligotrophic regions...". We will also note that "Iron is omitted from BFM17, limiting the applicability of the model in regions where iron components are important, such as the Southern Ocean and the tropical Pacific. Thus, if used in such a region, at least a fixed concentration of iron might be needed (though this has not been method has not been validated within BFM17)."

*Line 79: The "accuracy" certainly depends on the underlying measure. What is regarded as important will certainly depend on the selected region and the purpose of the model.*

We agree that "accuracy" is a relative term that depends on the intended goal of the model. We will thus adjust this statement in the revised paper to read: "It should be noted that the primary focus in the present study is to demonstrate the efficacy of using BFM17 as an accurate BGC model for high-resolution, high-fidelity upper-ocean process or parameterization studies, by comparing to results from observations and BFM56; as such, here we only consider one open-ocean location (i.e., the Sargasso Sea) and make the assumption that its ability to reproduce important and difficult key behaviors at this location, such as the initial spring bloom and subsequent subsurface chlorophyll maxima, are indications of its efficacy for use as a process study model."

*Line 83: So which parameters were calibrated and how was this done?*

We will provide in the revised manuscript an explanation of how the parameter values were determined. Briefly, the values used in this paper were based largely on the values established by Vichi *et al.* (2003, 2007, 2013) for BFM56. We are also currently pursuing computational optimization of these parameters to minimize model error when compared to the BATS observational data; this future work will be mentioned in the Conclusions section of the manuscript.

*Line 135: I agree that presenting all equations is important. Still I find it impossible to keep overview in the presented form. The rationale behind the construction of the new BFM17-model does not at all become clear to me.*

We agree with the reviewer that this section was rather long and confusing. We will move the detailed equations and 0D test to appendices at the end of the manuscript. This section will thus be focused on just the assumptions and choices of state variables for clarity with references to the appendices for further detail.

*Line 380: I find it very hard to understand the physical setup and would appreciate if*

*the main assumptions were summarized in the beginning of this Subsection. Currently, I get lost in too many details. Also, it should be clearly stated that only the upper 150m are simulated. What is the maximum mixed layer depth?*

We will summarize the physical setup assumptions and clearly state that only the upper 150m are simulated in the revised manuscript: "In order to focus on the upper-thermocline, open-ocean regime BFM17 was developed for, the physical model only extends 150 m in depth and diagnostically calculates diffusivity terms based upon proscribed temperature and salinity profiles from BATS. While a 1D physical model is unlikely to resolve all processes relevant for biogeochemistry in the upper-thermocline, we have made additions, such as large-scale general circulation and mesoscale eddy vertical velocities and relaxation bottom boundary conditions for nutrient upwelling, to better represent missing processes." Additionally, we will also state that the maximum mixed layer depth from the climatology is approximately 125 m based upon a $\Delta$ 0.02 kg/m$^3$ criteria in Section 4.2.

*Line 520: I would suggest to describe initial values and boundary conditions in relation to the description of the physical setup.*

Initial and boundary conditions are described in Section 4.4; the great length of the previous model description may have obscured these definitions, and by moving many of the model equations and 0D tests to the appendices, the specification of initial and boundary conditions should now be more readily apparent to the reader.

*Figure 2 What is the purpose of this figure? Why are not both model versions depicted and what do these experiments add to the conclusions?*

We again agree with the reviewer that differences in how the physics are represented in a 1D model can greatly affect the results. We thus include the 0D

[Figure]

test and this figure to show that, without a specific physical model, BFM17 reproduces a stable and well-behaved seasonal cycle similar to that observed in the BATS region. We will also include the following qualifications in the revised manuscript: "As an initial test of BFM17 (without the influence of any particular physical model) . . . " and ". . . indicating that a self-consistent and stable seasonal cycle with reasonable ecosystem values can be attained by the model, regardless of its coupling to a physical model." We do not include results from BFM56 in this figure as it is already a validated, established model [see the cited papers by Vichi *et al.* (2003, 2007, 2013)] and is not the focus of this manuscript. It should also be noted that this figure will be moved to an appendix at the end of the revised manuscript, de-emphasizing its overall importance, in line with the reviewer's comments.

*Figure 3+4: Again – the purpose of showing these figures does not become clear to me (e.g., Fig.4 vs. Fig.6?)*

Figure 3 shows the monthly input forcing profiles used in the 1D coupled model. Given that they are profiles derived here in this study (and thus not used in previous studies that could easily be referenced) and they are more easily described in figure format, rather than in text, we feel that Figure 3 is an important figure that should remain in the manuscript. Figure 4 shows the target climatological fields we aim to reproduce with the models. However, given that the top row of Figure 6 shows this same information, we agree that Figure 4 is redundant and will take it out of the revised manuscript.

*Figure 6: I see substantial differences between both model versions—even in the base currency "phosphate". How do the authors rate which differences matter? Why are the models so different from the observations at 150m (close to the bottom boundary conditions)?*

[Figure]

We again agree with the reviewer that the nutrient profiles deviate strongly from the observations at depth. This was due to some assumptions we made (described in detail in an earlier response) for a particular bottom boundary condition that we now feel are unsupported. We have changed this (again, details described in an earlier response) and the models now more closely reproduce the observations at depth, with improved correlation coefficients and *rms* errors. Additionally, we do not believe that we rate which differences matter, but rather present the results for all comparisons and allow the reader to make that decision for themselves based upon their intended application or study focus.

*Figure 7: Compared to the huge differences between the reference model and the observations, both model versions seem indeed rather similar (while in fact I regard these differences as substantial). I would suggest to add a more intuitive representation of the differences between both model versions in the corresponding text, e.g., % relative to the reference model.*

We agree with the reviewer that in comparison to the observations, the two models seem rather similar. To quantitatively support this conclusion, following the reviewer's suggestion we will present in the revised manuscript correlation coefficients and *rms* error values for BFM17 in comparison to BFM56 (similar to what was done for the BATS target fields) for each of the presented target fields. The correlations between the two models are as follows: Chlorophyll = 0.85, Oxygen = 0.56, Nitrate = 0.99, Phosphate = 0.99, PON = 0.95, NPP = 0.97.

*Table 8 + line 568: I like the idea behind this comparison to illustrate that the presented model is state-of-the-art. Still comparability needs to be ensured. Do all cited models refer to the same sampling in time (please specify) and state variables? Otherwise these numbers are not comparable. Also, the underlying physical model setups should be listed as well. It is much harder to simulate a specific timeseries in a 3D context than*

[Figure]

*to be able to tune the model at a single location (looking at Figure 1 and the tables, I would expect at least 40 poorly known parameters to match 12 moments in time – this should work perfectly at least for one state variable at one depth level).*

We agree with the reviewer that whether these models were comparable was not clearly stated. We will elaborate on each of the three models in the revised manuscript through the following additional text: "All models used climatological monthly mean forcing from the BATS region and reported climatological monthly means for their results. Care was taken to ensure that the same variable definition was compared between all models. Ayata *et al.* (2013) used a similar 1D physical model as was used here, while Spitz *et al.* (2001) and Fasham (1990) used a time-dependent box model of the upper-ocean mixed layer. As such, correlations and RMS error values for comparison to Ayata *et al.* (2013) are computed over the entire domain (Ayata *et al.* (2013) calculated their metrics over the top 168 m of their domain) and for comparison to Spitz *et al.* (2001) and Fasham (1990) were calculated only within the mixed layer (defined as the depth at which the density is 0.02 kg/m$^3$ greater than the surface density) and are shown as separate columns in Table 3."

*Line 605 (Conclusion): I would suggest to add a clear statement for which purpose the model BFM17 was designed and what it could be used for. Subsequently, the study would in my eyes benefit from an outline why the authors think that the model is up-to-the-task, or at least state-of-the-art.*

We agree that the original manuscript could have made this more clear. We now provide clear statements both at the beginning and in the conclusion of the revised manuscript that state this: "BFM17 was developed primarily for use within high-resolution, high-fidelity 3D physical models, such as LES, for process, parameterization, and optimal parameter estimation studies, an application for which its more complex counterpart BFM56 would be much too costly for." and "BFM17

is now of a size that it can be efficiently integrated in high-resolution, high-fidelity 3D simulations of the upper ocean, and future work will examine model results in this context"

*Line 615: As outlined before, I do not agree on the generalized, conclusive statement "the model does well".*

We agree that this statement was overly general, and so we will provide a more detailed and nuanced statement that the model accurately captures the subsurface chlorophyll maximum and bloom intensity observed in the BATS data, and that results from BFM17 are similar to those from BFM56.

---

## Author Response (AR1)

**Author Response to Reviews**

We appreciate the constructive and insightful comments from the reviewers. A number of changes have been made to the text at the reviewers' requests and we feel that the revised paper is now substantially clearer and stronger as a result. Detailed replies to each of the reviewers' points are provided below.

**Response to Reviewer 1**

*This manuscript concerns the development of a biogeochemical model of open-ocean ecosystem dynamics called BFM17. The authors make the case that the model is "complex and flexible enough to capture open-ocean ecosystem dynamics, but reduced enough to incorporate into highly resolved numerical simulations with limited additional computational cost." Furthermore, they provide validation of the model in 0D and 1D mode with measured data from an oligotrophic open ocean station. In general, I agree with the claims of the authors. My view is that reduced complexity models such as this one are important for process studies, so I welcome the initiative of the authors to develop one more option for biogeochemical modelers. It would be preferable to have a validation also in a 3D setting, which is the one that the model was developed for but, since the model equations use standard formulations for the source and sink terms, I believe that the validation presented in the manuscript is sufficient. Overall, I think the model has the potential to become a valuable addition to the tools available to study ocean biogeochemistry processes. With models such as these, only time can tell if it will fulfill this promise.*

**Comments:**

*My only remark is that, like similar papers, little time is dedicated to describe the way the parameter values are found. I find it peculiar that since, as everybody in this business knows, model parameters are paramount to getting decent model results, hardly no one takes the time to explain how they got to the values they are using. Thus, I urge the authors to add a little more detail on this matter, because it will certainly help other that may want to use the BFM17 model in their work.*

> We agree with the reviewer and we now provide an explanation of how the parameter values were determined on lines 128-129 and 131-133 of the revised manuscript: "The baseline parameters used in BFM17 are those detailed in Vichi et al. (2007)..." and "The only relevant difference with respect to Vichi et al. (2007) is related to the choice of the phytoplankton specific photosynthetic rate ($r_P^{(0)}$ in Table A3 of Appendix A); in this case, the new value was chosen according to the control laboratory cultures of Fiori et al. (2012)."

*Please check the reference style in line 347.*

> Thank you for catching this mistake. The correct reference style is now used in the revised manuscript.

**Response to Reviewer 2**

*The presented manuscript compares two model versions of a pelagic biogeochemical model, embedded in a 1-dimensional ocean model. The first model version refers to the BFM-model with 56 state variables (BFM56), while a second, reduced version (BFM17) includes only 17 state variables for the biogeochemistry. The authors compare the results of both model versions to climatological observations at the sites Bermuda Atlantic Timeseries (BATS) and Bermuda Testbed Mooring*

*(BTM).*

*The manuscript addresses the important, so far unresolved question on the necessary complexity of pelagic biogeochemical models. The topic is per se interesting. Still I have some major concerns.*

**Major Comments:**

*BFM17 excludes some processes relative to BFM56. This reduces by construction the applicability of the newly developed model (e.g., coastal shallow regions might not be represented so well anymore, due to the lack of the sediment processes, while simulating global phosphate distributions might still work very well). I would thus appreciate a clear statement for which purposes the new model was designed.*

> We agree with the reviewer that the applicability of BFM17 is reduced in comparison to BFM56 and, in order to more clearly state the range of applicability of the model, we state in the revised manuscript on line 115 that "BFM17 is a pelagic model intended for oligotrophic regions that are not iron or silicate limited...". With respect to the purpose of the model, we clarify on lines 117-120 that "We have developed BFM17 primarily for use with high-resolution, high-fidelity numerical simulations, including large eddy simulations (LES) used in process, parameterization, and parameter optimization studies. As such, we do not validate the efficacy of BFM17 as a global BGC model, and note that it is missing potentially important processes for such an application, which we elaborate on shortly."

*Also, the overall aim of the presented results did not get entirely clear to me: is the idea of BMF17 to mimic/resemble certain processes of BFM56 or is BFM17 a model on its own, which might perform even better than BFM56 relative to the observations?*

> We agree with the reviewer that this was not entirely clear and we have added the following text to the revised manuscript on lines 120-123: "We also note that we compare BFM17 to the original BFM56 in Section 5 to demonstrate that, although it is reduced in complexity, BFM17 is equally appropriate for use in seasonal process, parameterization, and optimal parameter estimation studies for which a more complex model such as BFM56 may be too computationally expensive."

*In my eyes the study would need a better illustration that the presented model is up to the tasks it is designed for–or at least state-of-the-art. Currently, both model versions are compared to climatological, station based in-situ observations close to Bermuda. I am surprised that both model versions dont do better (e.g., for oxygen the explained variances are in the order of 5-15%, cf. Fig.7). The study might get more convincing if the authors could at least outline what goes wrong.*

> We agree with the reviewer that not all presented fields are reproduced by BFM17 and BFM56 with great accuracy (e.g., oxygen), and we now clarify that the exact reproduction of all fields was not our goal with BFM17. In the revised manuscript, we state on lines 345-353 that "...oxygen is historically difficult to predict using BGC models of any complexity. It is likely that this is due, in part, to inaccuracies in the mixing parameterizations used in POM 1D and other physical models. For example, BFM17 struggles to accurately predict oxygen, in part, because the second-order mixing scheme of Mellor and Yamada (1982) lacks sufficient resolution of the winter mixing using just the monthly mean temperature and salinity. However, since it is often not included or presented at all in models of similar complexity to

BFM17 (i.e., models reduced enough to reasonably couple to a high-fidelity, high-resolution physical model), studies that explore this hypothesis have been difficult to undertake. Thus, we include oxygen in BFM17 and present our results here to illustrate this exact point, and to lend motivation to developing and using a model such as BFM17 to study the effects of physical processes missing from mixing parameterizations and how they can be better represented."

*When reading the model description, I found it rather difficult to keep overview what exactly was done. I would suggest to summarize the main model assumptions somewhere in the beginning of the model description (the equations might as well be collectively listed in an appendix). E.g. it did not become clear to me how the reduction of state variables was motivated and how the new state variables and parameter settings were chosen.*

We agree that our previous model description was potentially difficult to follow. We have thus moved the detailed equations and 0D test to appendices in the revised manuscript (now lines 440-686). This now helps to focus the reader in the main model description section on the model assumptions and choice of state variables.

*I agree in general with the approach to compare both model versions initially in a low-dimensional environment, which is less costly to run than a full-fledged ocean model. Unfortunately, the authors do not take advantage of such a computationally cheap setup, e.g., by testing the sensitivities of both models towards environmental changes in idealized experiments. Such an approach would be feasible without too much effort and could explore how much BFM17 resembles BFM56 under various conditions (preferably also in well fertilized regions).*

We agree with the reviewer that this kind of comparison would be computationally cheap and easy to perform. However, since BFM17 was primarily developed for use in high-resolution, high-fidelity physical models (something BFM56 cannot be used for because of its cost) and not for use as a global or column model, this comparison is beyond the scope of the model description and baseline demonstration that are the primary objectives of the present paper. Nevertheless, we do agree with the reviewer that this is an important direction for future research, and we now clarify this point in the revised manuscript with the following text on lines 81-89: "It should be noted that the primary focus of the present study is to demonstrate the viability of BFM17 as an accurate BGC model for high-resolution, high-fidelity simulations of the upper ocean used in process, parameterization, and parameter optimization studies. This is accomplished here by comparing results from BFM17 to results from observations and BFM56; as such, here we only consider one open-ocean location (i.e., the Sargasso Sea). Although the model must also be applied at other locations to determine its general applicability, its ability, demonstrated herein, to reproduce important and difficult key behaviors in the Sargasso Sea, such as the initial spring bloom and subsequent subsurface chlorophyll maxima, supports its use as a process study model. The correspondence between BFM17 and the more general BFM56 also provides confidence that the reduced model will prove effective at modeling other ocean locations and conditions, and exploring the range of applicability of BFM17 remains an important direction for future research."

*I am, however, sceptic about comparing the 1-dimensional model results to the observations. First, the comparison captures only one very special region while I assume that the model is rather designed to answer questions on a global scale and I regard a 3-dimensional ocean component as*

We apologise for any confusion we might have caused by not clearly stating that BFM17 was not developed for use as a global model. We now clearly state this in the revised manuscript on lines 117-120: "We have developed BFM17 primarily for use with high-resolution, high-fidelity numerical simulations, including large eddy simulations (LES) used in process, parameterization, and parameter optimization studies. As such, we do not validate the efficacy of BFM17 as a global BGC model, and note that it is missing potentially important processes for such an application, which we elaborate on shortly."

*Secondly, I fear that ocean mixing might not be realistically represented and that the restoring below 150m might induce spurious effects in the 1-dimensional model. In the real ocean, regions with intense sinking of organic matter and upwelling of nutrients are often spatially decoupled. How I understood the presented setup, the 1-dimensional model needs to balance both processes in a single box and thus induces some eddy driven transports (line 504 + line 515), to reach some kind of steady-state (line 526). The latter seems rather arbitrary to me (please correct me in case I got things wrong). My scepticism is supported by the rather poor representation of oxygen in both model versions (because oceanic oxygen concentrations close to the surface are strongly impacted by water temperature and mixing).*

We agree with the reviewer that any 1D column model is unlikely to realistically represent all processes, particularly those associated with mixing and spatially decoupled processes. However, the 1D physical model is only used here for validation purposes, and a primary reason for developing BFM17 is to explore issues such as the effects of mixing and temperature in the surface ocean on oxygen or how "patchy" (or spatially decoupled) nutrient upwelling effects the distribution and fate of a phytoplankton bloom using more physically realistic fully non-hydrostatic 3D simulations. We now elaborate on this in the text of the revised manuscript through the following additions on lines 135-144: "Dissolved oxygen is of particular interest, because it is historically difficult to predict using BGC models of any complexity. This is likely due, in part, to missing physical processes in the mixing parameterizations used in global and column models. This provides motivation for the present study, since a primary goal in the development of BFM17 is to create a BGC model that can be used in combination with high-resolution, high-fidelity physical models (e.g., those found in LES) to understand the effects of these physical processes and how they can be more accurately represented in mixing parameterizations. Dissolved and particulate organic matter, each with their own partitions of carbon, nitrogen, and phosphate, are also included in BFM17 to account for nutrient recycling and carbon export due to particle sinking. Another primary goal of developing BFM17 is to explore how spatially decoupled (or patchy) processes, such as the sinking of organic matter and the subsequent upwelling of multiple recycled nutrients (not just nitrate) affect the fate and distribution of a phytoplankton bloom."

*Also, I am puzzled that both models deviate strongly from the observations even at 150m i.e. close to the prescribed boundary conditions at depth (cf., Fig. 6).*

We agree with the reviewer that the nutrient profiles (particularly nitrate) deviate strongly from the observations at depth. Initially, in order to represent the recycling of PON and surface layer nitrogen replenishment from depth, we included a bottom boundary condition

for ammonium based on the gradient in PON observations (direct observations of ammonium at the BATS location were lacking). However, as the reviewer points out, this process can be spatially decoupled and thus the assumptions we made associated with this boundary condition are potentially unsupported. In light of the fact that we do not know the exact relationship between the sinking PON and up-welled ammonium, we have now chosen to set this boundary condition to zero (consistent with the idea that most of the nitrogen flux from depth is occurring as nitrate flux). The nutrient results for both models now more closely match observations (see Figure 4 of the revised manuscript), both overall and at depth. The new correlation coefficients and *rms* errors in parentheses are now as follows for BFM17 (and can be found in Table 3 of the revised manuscript): Chlorophyll = 0.63 (0.08), Oxygen = 0.37 (31.18), Nitrate = 0.94 (0.22), Phosphate = 0.91 (0.01), PON = 0.85 (0.15), NPP = 0.93 (0.26). For BFM56, they are: Chlorophyll = 0.60 (0.10), Oxygen = 0.18 (21.84), Nitrate = 0.93 (0.16), Phosphate = 0.93 (0.005), PON = 0.85 (0.11), NPP = 0.87 (0.63).

**Minor Comments:**

*Line 1: I find the expression "reduced-order model" in the presented context a bit confusing. The authors seem to use this term whenever a certain (non-defined) number of state variables is undercut. It might make sense to define the expression somewhere.*

We agree that this expression is confusing. We have removed this description and replaced it with "upper-thermocline, open-ocean" throughout the revised manuscript in order to be clear as to how exactly the model is "reduced".

*Line 7: I expect large differences in the tropical Pacific when disregarding iron in BFM17 (unless the top-down control is too strong). Depending on the region of interest the authors might consider to use at least a fixed, prescribed iron concentrations.*

We agree with the reviewer that BFM17 may not perform well in regions where iron-limitation is important, such as the tropical Pacific and the Southern Ocean. We believe that the following altered statements on lines 6-7 and 115 in the revised manuscript convey that BFM17 was not developed, nor has it been qualified, for used in such regions: "To reduce the overall computational cost and to focus on upper-thermocline, open-ocean, and non-iron or silicate limited conditions,..." and "BFM17 is a pelagic model intended for oligotrophic regions that are not iron or silicate limited...". We also note on lines 148-150 that "Iron is omitted from BFM17, limiting the applicability of the model in regions where iron components are important, such as the Southern Ocean and the tropical Pacific. Thus, if used in such regions, at least a fixed concentration of iron may be needed (although this method has not yet been validated within BFM17)."

*Line 79: The "accuracy" certainly depends on the underlying measure. What is regarded as important will certainly depend on the selected region and the purpose of the model.*

We agree that "accuracy" is a relative term that depends on the intended goal of the model. We have thus adjusted this statement on lines 81-87 in the revised paper to read: "It should be noted that the primary focus of the present study is to demonstrate the viability of BFM17 as an accurate BGC model for high-resolution, high-fidelity simulations of the upper ocean used in process, parameterization, and parameter optimization studies. This is accomplished here

by comparing results from BFM17 to results from observations and BFM56; as such, here we only consider one open-ocean location (i.e., the Sargasso Sea). Although the model must also be applied at other locations to determine its general applicability, its ability, demonstrated herein, to reproduce important and difficult key behaviors in the Sargasso Sea, such as the initial spring bloom and subsequent subsurface chlorophyll maxima, supports its use as a process study model."

*Line 83: So which parameters were calibrated and how was this done?*

We now provide an explanation of how the parameter values were determined on lines 128-129 and 131-133 of the revised manuscript: "The baseline parameters used in BFM17 are those detailed in Vichi et al. (2007)..." and "The only relevant difference with respect to Vichi et al. (2007) is related to the choice of the phytoplankton specific photosynthetic rate ($r_P^{(0)}$ in Table A3 of Appendix A); in this case, the new value was chosen according to the control laboratory cultures of Fiori et al. (2012)."

*Line 135: I agree that presenting all equations is important. Still I find it impossible to keep overview in the presented form. The rationale behind the construction of the new BFM17-model does not at all become clear to me.*

We agree with the reviewer that this section was rather long and confusing. We have now moved the detailed equations and 0D test to appendices at the end of the manuscript (now lines 440-686). This section now focuses on just the assumptions and choices of state variables for clarity with references to the appendices for further detail.

*Line 380: I find it very hard to understand the physical setup and would appreciate if the main assumptions were summarized in the beginning of this Subsection. Currently, I get lost in too many details. Also, it should be clearly stated that only the upper 150m are simulated. What is the maximum mixed layer depth?*

We now summarize the physical setup assumptions and clearly state that only the upper 150m are simulated in the revised manuscript on lines 168-173: "In order to focus on the upper-thermocline, open-ocean regime for which BFM17 was developed, the physical model only extends 150 m in depth and diagnostically calculates diffusivity terms based upon prescribed temperature and salinity profiles from the observations. While a 1D physical model is unlikely to resolve all processes relevant for biogeochemistry in the upper thermocline, we have made additions, such as large-scale general circulation and mesoscale eddy vertical velocities, as well as relaxation bottom boundary conditions for nutrient upwelling, to better represent missing processes." Additionally, we also state on lines 288-289 of the revised manuscript that the "maximum mixed layer depth from the climatology is approximately 125 m based upon a $\Delta$ 0.02 kg/m$^3$ criteria."

*Line 520: I would suggest to describe initial values and boundary conditions in relation to the description of the physical setup.*

Initial and boundary conditions are described in Section 4.4; the great length of the previous model description may have obscured these definitions, and by moving many of the model equations and 0D tests to the appendices, the specification of initial and boundary conditions

should now be more readily apparent to the reader.

*Figure 2 What is the purpose of this figure? Why are not both model versions depicted and what do these experiments add to the conclusions?*

We again agree with the reviewer that differences in how the physics are represented in a 1D model can greatly affect the results. We thus include the 0D test and this figure to show that, without a specific physical model, BFM17 reproduces a stable and well-behaved seasonal cycle similar to that observed in the BATS region. We now also include the following qualifications in the revised manuscript on lines 664 and 682-683: "As a simple test of BFM17 without the influence of any particular physical model…" and "…indicating that a self-consistent and stable seasonal cycle with reasonable ecosystem values can be attained by the model, regardless of its coupling to a physical model." We do not include results from BFM56 in this figure as it is already a validated, established model [see the cited papers by Vichi *et al.* (2003, 2007, 2013)] and is not the focus of this manuscript. It should also be noted that this figure has been moved to an appendix at the end of the revised manuscript, de-emphasizing its overall importance, in line with the reviewer's comments.

*Figure 3+4: Again the purpose of showing these figures does not become clear to me (e.g., Fig.4 vs. Fig.6?)*

Figure 3 shows the monthly input forcing profiles used in the 1D coupled model. Given that they are profiles derived here in this study (and thus not used in previous studies that could easily be referenced) and they are more easily described in figure format, rather than in text, we feel that Figure 3 is an important figure that should remain in the manuscript. Figure 4 shows the target climatological fields we aim to reproduce with the models. However, given that the top row of Figure 6 (now Figure 4 in the revised manuscript) shows this same information, we agree that Figure 4 is redundant and have taken it out of the revised manuscript.

*Figure 6: I see substantial differences between both model versions—even in the base currency "phosphate". How do the authors rate which differences matter? Why are the models so different from the observations at 150m (close to the bottom boundary conditions)?*

We again agree with the reviewer that the nutrient profiles deviate strongly from the observations at depth. This was due to some assumptions we made (described in detail in an earlier response) for a particular bottom boundary condition that we now feel are unsupported. We have changed this (again, details described in an earlier response) and the models now more closely reproduce the observations at depth, with improved correlation coefficients and *rms* errors. Additionally, we do not believe that we rate which differences matter, but rather present the results for all comparisons and allow the reader to make that decision for themselves based upon their intended application or study focus.

*Figure 7: Compared to the huge differences between the reference model and the observations, both model versions seem indeed rather similar (while in fact I regard these differences as substantial). I would suggest to add a more intuitive representation of the differences between both model versions in the corresponding text, e.g., % relative to the reference model.*

We agree with the reviewer that in comparison to the observations, the two models seem

rather similar. To quantitatively support this conclusion, following the reviewer's suggestion we now present on lines 343-344 in the revised manuscript correlation coefficients and *rms* error values for BFM17 in comparison to BFM56 (similar to what was done for the BATS target fields) for each of the presented target fields:"Correlation coefficients between the two models are 0.85 for chlorophyll, 0.56 for oxygen, 0.99 for nitrate, 0.99 for phosphate, 0.95 for PON, and 0.97 for NPP."

*Table 8 + line 568: I like the idea behind this comparison to illustrate that the presented model is state-of-the-art. Still comparability needs to be ensured. Do all cited models refer to the same sampling in time (please specify) and state variables? Otherwise these numbers are not comparable. Also, the underlying physical model setups should be listed as well. It is much harder to simulate a specific timeseries in a 3D context than to be able to tune the model at a single location (looking at Figure 1 and the tables, I would expect at least 40 poorly known parameters to match 12 moments in time  this should work perfectly at least for one state variable at one depth level).*

We agree with the reviewer that whether these models were comparable was not clearly stated. We now elaborate on each of the three models in the revised manuscript through the following additional text on lines 378-384: "All models used climatological monthly mean forcing from the BATS region and reported climatological monthly means for their results. Care was taken to ensure that the same variable definition was compared between all models. Ayata et al. (2013) used a similar 1D physical model as was used here, while Spitz et al. (2001) and Fasham et al. (1990) used a time-dependent box model of the upper-ocean mixed layer. As such, correlations and RMS error values for comparison to Ayata et al. (2013) were computed over the entire domain (Ayata et al. (2013) calculated their metrics over the top 168 m of their domain). For comparison to Spitz et al. (2001) and Fasham et al. (1990), correlations and *rms* errors were calculated only within the mixed layer (defined as the depth at which the density is 0.02 kg/m$^3$ greater than the surface density) and are shown as separate columns in Table 3."

*Line 605 (Conclusion): I would suggest to add a clear statement for which purpose the model BFM17 was designed and what it could be used for. Subsequently, the study would in my eyes benefit from an outline why the authors think that the model is up-to-the-task, or at least state-of-the-art.*

We agree that the original manuscript could have made this more clear. We now provide clear statements both at the beginning and in the conclusion of the revised manuscript that state this (lines 117-119, 423-425, and 437-438): "We have developed BFM17 primarily for use with high-resolution, high-fidelity numerical simulations, including large eddy simulations (LES) used in process, parameterization, and parameter optimization studies", "BFM17 was developed primarily for use within high-resolution, high-fidelity 3D physical models, such as LES, for process, parameterization, and parameter optimization studies, applications for which its more complex counterpart BFM56 would be much too costly", and "BFM17 is now of a size that it can be efficiently integrated in high-resolution, high-fidelity 3D simulations of the upper ocean, and future work will examine model results in this context."

*Line 615: As outlined before, I do not agree on the generalized, conclusive statement "the model does well".*

We agree that this statement was overly general, and so we have provided a more detailed

[revised manuscript text omitted]

2. Phytoplankton functional group in the living organic CFF, nitrogen constituent (state variable $P_N$):

$$\frac{\partial P_N}{\partial t}\bigg|_{bio} = \max\left[0, \frac{\partial P_N}{\partial t}\bigg|^{upt}_{N^{(2)}} + \frac{\partial P_N}{\partial t}\bigg|^{upt}_{N^{(3)}}\right] - \frac{\partial P_N}{\partial t}\bigg|^{lys}_{R_N^{(1)}} - \frac{\partial P_N}{\partial t}\bigg|^{lys}_{R_N^{(2)}} - \frac{\partial P_N}{\partial t}\bigg|^{prd}_{Z_N}, \tag{A6}$$

3. Phytoplankton functional group in the living organic CFF, phosphorus constituent (state variable $P_P$):

$$\frac{\partial P_P}{\partial t}\bigg|_{bio} = \max\left[0, \frac{\partial P_P}{\partial t}\bigg|^{upt}_{N^{(1)}}\right] - \frac{\partial P_P}{\partial t}\bigg|^{lys}_{R_P^{(1)}} - \frac{\partial P_P}{\partial t}\bigg|^{lys}_{R_P^{(2)}} - \frac{\partial P_P}{\partial t}\bigg|^{prd}_{Z_P}, \tag{A7}$$

4. Phytoplankton functional group in the living organic CFF, chlorophyll constituent (state variable $P_{chl}$):

$$\frac{\partial P_{chl}}{\partial t}\bigg|_{bio} = \frac{\partial P_{chl}}{\partial t}\bigg|^{syn} - \frac{\partial P_{chl}}{\partial t}\bigg|^{loss}, \tag{A8}$$

**Table A1.** Symbols, values, units, and descriptions for environmental parameters within the BFM17 pelagic model.

| Symbol | Value | Units | Description |
|---|---|---|---|
| $Q_{10,P}$ | 2.00 | - | Phytoplankton $Q_{10}$ coefficient |
| $Q_{10,Z}$ | 2.00 | - | Zooplankton $Q_{10}$ coefficient |
| $Q_{10,N}$ | 2.00 | - | Nitrification $Q_{10}$ coefficient |
| $T^*$ | 10.0 | °C | Base temperature |
| $c_P$ | 0.03 | $m^2$ (mg chl)$^{-1}$ | Chlorophyll-specific light absorption coefficient |
| $\varepsilon_{PAR}$ | 0.40 | - | Fraction of photosynthetically active radiation |
| $\lambda_w$ | 0.0435 | $m^{-1}$ | Background attenuation coefficient |
| $c_{R^{(2)}}$ | $0.1 \times 10^{-3}$ | $m^2$ (mg C)$^{-1}$ | C-specific attenuation coefficient of particulate detritus |

**Table A2.** List of abbreviations used to indicate physiological and ecological processes in the equations comprising the BFM17 pelagic model.

| Abbreviation | Process |
|---|---|
| gpp | Gross primary production |
| rsp | Respiration |
| prd | Predation |
| rel | Biological release: egestion, excretion, mortality |
| exu | Exudation |
| upt | Uptake |
| lys | Lysis |
| syn | Biochemical synthesis |
| loss | Biochemical loss |
| nit | Nitrification |

785 where the descriptions of each of the source and sink terms are provided in Table A2. The subscript "bio" on the left-hand side terms indicates that these are the total rate expressions associated with all biological processes.

For the evolution of the phytoplankton carbon constituent given by Eq. (A5), gross primary production depends on the non-dimensional regulation factors for temperature and light as well as on the maximum photosynthetic growth rate and the phytoplankton carbon instantaneous concentration. This then gives

790
$$\left.\frac{\partial P_C}{\partial t}\right|_{CO_2}^{gpp} = r_P^{(0)} f_P^{(T)} f_P^{(E)} P_C, \tag{A9}$$

where $r_P^{(0)}$ is the maximum photosynthetic rate for phytoplankton (reported in Table A3) and $f_P^{(T)}$ is the temperature regulation factor for phytoplankton given by Eq. (A1). The term $f_P^{(E)}$ is the light regulation factor for phytoplankton, which is defined following (Jassby and Platt, 1976) as

$$f_P^{(E)} = 1 - \exp\left(-\frac{E_{PAR}}{E_K}\right), \tag{A10}$$

795 where $E_{PAR}$ is defined in Eq. (A3) and $E_K$ (the "optimal" irradiance) is given by

$$E_K = \left[\frac{r_P^{(0)}}{\alpha_{chl}^{(0)}}\right]\left(\frac{P_C}{P_{chl}}\right). \tag{A11}$$

The parameter $\alpha_{chl}^{(0)}$ is the maximum light utilization coefficient and is defined in Table A3.

**Table A3.** Phytoplankton parameters, values, units, and descriptions within the BFM17 pelagic model.

[revised manuscript text omitted]

---

## Author Response (AR2)

**To the Editor**

We appreciate the constructive comments from the reviewer, and we have completely agreed with and addressed each of these comments, as outlined below. Through these changes, we feel that the revised paper is now substantially clearer and stronger. Detailed replies to each of the reviewer's points are provided below.

*Page 1, Line 12: it would be nice to see some example numbers which illustrate what is meant by good*

We now include correlation coefficients to illustrate this.

*Page 1, Line 14ff: According to Table 3 the judgement of the model performances depends on the considered variable and also all the models are not directly comparable. Also, the considered Fasham-model is now more than 30 years old. I would thus suggest to tone down. I am fine with a statement that the performance of the new model at the test location is well within the range of comparable models. Also, some specific strengths of the model could be listed in addition.*

We have toned down this statement and added some strengths.

*Page 2, Line 41: I find this difficult to read (unclear what "those" and "they" refer to).*

We now clarify what is being referred to.

*Page 2, Line 46ff: Split the sentence to make it easier to read?*

We have split the sentence into two.

*Page 2, Line 50 Better? "…, a second common BGC modelling approach…"*

We have made this change.

*Page2, Line 59: the disconnection*

We have changed this word to "difference" instead of "disconnect."

*Page 3, Line 71: I would suggest to start a new sentence after "ratios".*

We have made this change.

*Page 3, Line 79: obtaining → deriving*

We have made this change.

*Page3, Line 81: I would rather say that the focus is to introduce the new model and to provide a first rough comparisons to BFM56 and observations, because the model assessment in the presented manuscript is rather limited.*

We have made this change.

*Page 3, Line 84ff: This sentence is rather long and difficult to follow. What is the key message*

*here?*

We have cleaned up this sentence to help clarify its message.

*Page 3, Line 100ff. I would be more careful with this statement (because it is not proven) and use "we anticipate ... that ... might ..." instead of "postulate..."*

We have made this change.

*Page4, Line 116: lesser significance → minor importance*

We have made this change.

*Page 4, Line 117ff: I find "subject to the constraint" difficult to understand and would also suggest to start a new sentence here.*

We have removed this wording and started a new sentence here to provide clarity.

*Page 4, Line 150ff: I find this a bit confusing because top-down control can act at different levels.*

We have now added to this sentence that the top-down control we are referring to is an explicit term for predation of zooplankton.

*Page 7, Line 176 The notation of this equation is rather unusual. Also, the variables and abbreviations should be introduced directly (e.g., W appears first in line 187). The same holds for most other equations in this section.*

We now clarify that all equations in this section and in Appendix A use this notation because it is the same as it used in all other BFM papers, for consistency. Additionally, we now introduce all variables directly after they are used.

*Page 7, Line 194ff: It should be mentioned that only the upper part of the water column is simulated (please provide the respective depth level).*

We have now indicated in this sentence that we only simulate the upper 150m of the water column.

*Page 7, Line 200: It does not become clear where the prescribed temperatures and salinities originate from observations or the 3D model?*

We now mention within this sentence that the temperatures and salinities are obtained from the observations.

*Page 7, Line 201: How are the horizontal velocities obtained and how are they used in the coupled setup?*

We now mention within this sentence that the horizontal velocities are from the POM 1D physical model.

*Page 9-10: I must admit that I am lost.*

We now preface this section of text with a sentence highlighting the fact that the ensuing discussion is a description of the equations solved in the POM-1D physical model.

*Page 11, Line 276: Its problematic to use the same data for calibration and validation. I would rather suggest to refer to a first model test here.*

We have replaced "calibration and validation" with "testing".

*Page 11, Line 285: Please specify how the smoothing was done.*

We now specify here that it was done with a locally estimated scatterplot smoothing (LOESS) method.

*Page 12, Line 289: I missed what this criterion is used for.*

We have added clarification to this sentence to make it clear that this criteria was used to determine the maximum mixed layer depth in the climatology.

*Page 12, Line 290: What is meant by "similar processing"?*

We now clarify that this is referring to the same averaging, interpolating, and smoothing process described for the physical variables.

*Page 12, Fig.2: Could the mixed layer depths be included?*

We have now included the mixed layer depths in Fig. 2, panel (a).

*Page 13, Line 298: Could it be briefly described what the correction does?*

We now specify that this correction specifically reduces the error incurred by linearly interpolating the monthly averages to the model time step.

*Page 13, Line 325: How was this splitting done?*

We now specify that the the total initial carbon values for phytoplankton and zooplankton used in BFM17 were split into equal amounts for all the phytoplankton and zooplankton groups in BFM56.

*Page 14, Line 331: I would not call this validation  rather "model assessment at a test location" or similar.*

We have changed the section title to "Model Assessment Results".

*Page 14, Line 349ff: I don't see that the simulated subsurface chlorophyll maximum is particularly pronounced in summer. But maybe this refers to the color scale?*

We now specify in parentheses at the end of this sentence that this maximum is evident in Figure 4(g) as a larger chlorophyll concentration at depths close to 100m during the summer months.

*Page 14, Line 341ff: I disagree that the oxygen fields look similar. Adding correlations might,*

*however, convince me. Otherwise, I agree with the authors that the large discrepancy is very likely related to the representation of mixing in 1D and I think there is no drawback just to mention this here (as the authors do anyway later in the text and which could be removed then).*

We agree that this statement was too strong and have changed it to "…not completely dissimilar…"

*Page 14, Line 342ff: I would not talk about a trend in a seasonal cycle.*

We now state that "These results are consistent with those from BFM56…"

*Page 14, Line 344: Is there a potential reason why there are discrepancies in chlorophyll and oxygen?*

We now offer potential reasons for these discrepancies, namely, the additional phytoplankton and zooplankton groups, nitrification, and explicit representation of remineralization in BFM56.

*Page 15, Line 354: "Talk about quantitative evaluation" and "skill assessment" is in my eyes a bit too much - due to the limitation to a single location in an 1d model environment, I would rather call this "a first indication on the model performance".*

We have made this change.

*Page 17, Line 386ff: A lot of these findings were already described earlier.*

We have removed this paragraph.

*Page 17, Line 400: I do not agree here because the model performances depend on the considered variables (as also outlined in the manuscript above).*

We have now scaled this claim back to not include all of the target fields.

*Page 21f, Line 444: Does the BG model feedback on temperature and irradiance? If not this sentence is confusing. Also, I would tend to delete "inputs".*

We now clarify that BFM17 is only influenced by the environment through the temperature and irradiance and have deleted "inputs".

*Page 21ff, Appendix A2: I find it very difficult to follow the equations. Fist the notation is very unusual and then generally the abbreviations are not introduced directly. Partly I can kind of guess what it meant.*

We now clarify that all equations in Appendix A adopt this notation because it is the same as it used in all other BFM papers, for consistency. We introduce the abbreviations in Table A2, for clarity.

*Page 31ff, Appendix B: The idea behind constructing a sinusoidal forcing does not get clear to me and also I missed how winter and summer extremes were defined. Why did the authors not consider measurements here? It might make sense to skip B1, unless there is a clear purpose behind it.*

We now clarify that we used an idealized version of the observations for simplicity, as there is no

physical dimension in the 0D framework to properly apply the observational variables to.

*Line 682: What is meant by "self-consistent"?*

We have removed this term to reduce confusion.

*Line 685ff: I did not gain any confidence in the accuracy of the model by Fig. B1. Please rephrase and make the idea behind these experiments more clear.*

We have added, at the beginning of Appendix B, that the idea behind this test is to confirm its efficacy as viable BGC model without the – sometimes heavy – influence of any one particular physical model.